# Role of carbonate burial in Blue Carbon budgets

V. Saderne [1], N.R. Geraldi [1], P.I. Macreadie[2], D.T. Maher [3], J.J. Middelburg [4], O. Serrano [5], H. Almahasheer [6], A. Arias-Ortiz [7], M. Cusack[1], B.D. Eyre[8], J.W. Fourqurean [9], H. Kennedy[10], D. Krause-Jensen[11,12], T. Kuwae[13], P.S. Lavery[5], C.E. Lovelock[14], N. Marba [15], P. Masqué [5,7,16], M.A. Mateo[5,17], I. Mazarrasa[18], K.J. McGlathery[19], M.P.J. Oreska[19], C.J. Sanders[20], I.R. Santos [20], J.M. Smoak [21], T. Tanaya[13], K. Watanabe [13] & C.M. Duarte [1]

Calcium carbonates ($CaCO_3$) often accumulate in mangrove and seagrass sediments. As $CaCO_3$ production emits $CO_2$, there is concern that this may partially offset the role of Blue Carbon ecosystems as $CO_2$ sinks through the burial of organic carbon ($C_{org}$). A global collection of data on inorganic carbon burial rates ($C_{inorg}$, 12% of $CaCO_3$ mass) revealed global rates of 0.8 $TgC_{inorg}$ $yr^{-1}$ and 15–62 $TgC_{inorg}$ $yr^{-1}$ in mangrove and seagrass ecosystems, respectively. In seagrass, $CaCO_3$ burial may correspond to an offset of 30% of the net $CO_2$ sequestration. However, a mass balance assessment highlights that the $C_{inorg}$ burial is mainly supported by inputs from adjacent ecosystems rather than by local calcification, and that Blue Carbon ecosystems are sites of net $CaCO_3$ dissolution. Hence, $CaCO_3$ burial in Blue Carbon ecosystems contribute to seabed elevation and therefore buffers sea-level rise, without undermining their role as $CO_2$ sinks.

[1] King Abdullah University of Science and Technology (KAUST), Red Sea (RSRC) and Computational Bioscience (CBRC) Research Centers, Thuwal 23955-6900, Saudi Arabia. [2] School of Life and Environmental Sciences, Centre for Integrative Ecology, Deakin University, Geelong, Victoria 3216, Australia. [3] Southern Cross Geoscience, Southern Cross University, Lismore, New South Wales 2480, Australia. [4] Department of Earth Sciences, Utrecht University, Vening Meineszgebouw A, Princetonlaan 8a, Utrecht 3584, The Netherlands. [5] School of Science and Centre for Marine Ecosystems Research, Edith Cowan University, 270 Joondalup Drive, Joondalup, West Australia 6027, Australia. [6] Department of Biology, College of Science, Imam Abdulrahman Bin Faisal University (IAU), Dammam 31441-1982, Saudi Arabia. [7] Institut de Ciència i Tecnologia Ambientals, Universitat Autònoma de Barcelona, Bellaterra, Barcelona 08193, Spain. [8] Centre for Coastal Biogeochemistry, School of Environment, Science and Engineering, Southern Cross University, Lismore, New South Wales 2480, Australia. [9] Department of Biological Sciences, Center for Coastal oceans Research, Institute of Water and Environment, Florida International University, Miami 11200 SW 8th Street, Miami, Florida 33199, USA. [10] School of Ocean Sciences, Bangor University, Menai Bridge, Anglesey, Wales LL59 5AB, UK. [11] Department of Bioscience, Aarhus University, Vejlsøvej 25, Silkeborg 8600, Denmark. [12] Arctic Research Centre, Department of Bioscience, Aarhus University, Ny Munkegade 114, Building 1540, Århus C 8000, Denmark. [13] Coastal and Estuarine Environment Research Group, Port and Airport Research Institute, 3-1-1 Nagase, Yokosuka 239-0826, Japan. [14] School of Biological Sciences, The University of Queensland, St Lucia, Brisbane, Queensland 4072, Australia. [15] Department of Global Change Research, IMEDEA (CSIC-UIB), Institut Mediterrani d'Estudis Avançats Miquel Marquès 21, Esporles (Illes Balears) 07190, Spain. [16] Oceans Institute and School of Physics, University of Western Australia, 35 Stirling Highway, Crawley, West Australia 6009, Australia. [17] Centro de Estudios Avanzados de Blanes, Consejo Superior de Investigaciones Cientificas, Blanes 17300, Spain. [18] Environmental Hydraulics Institute "IH Cantabria", C/Isabel Torres No 15, Parque Científico y Tecnológico de Cantabria, Universidad de Cantabria, Santander 39011, Spain. [19] Department of Environmental Sciences, University of Virginia, Charlottesville, Virginia 22904, USA. [20] National Marine Science Centre, School of Environment, Science and Engineering, Southern Cross University, Cos Harbour, New South Wales 2450, Australia. [21] University of South Florida, St. Petersburg, Florida 33701, USA. Correspondence and requests for materials should be addressed to V.S. (email: vincent.saderne@kaust.edu.sa)

Mangrove forests and seagrass meadows have the capacity to elevate the seabed through the accretion of inorganic and organic particles[1] at global rates of ~0.5 and ~0.2 cm yr$^{-1}$, respectively[1]. Sediment accretion in mangrove forests and seagrass meadows leads to the sequestration of organic carbon ($C_{org}$)[2,3] originating from within and outside of the vegetated ecosystem[4]. Although mangroves and seagrass ecosystems occupy only a small fraction of the total coastal area (< 2%), they contribute 10% and 25% to the yearly $C_{org}$ sequestration in the coastal zone[1,5], respectively. Recognition of mangrove and seagrass meadows, together with saltmarshes, as sites of intense $C_{org}$ burial led to the formulation of Blue Carbon strategies to mitigate and adapt to climate change, through conservation and restoration of these ecosystems[1,6–8]. The focus on Blue Carbon has provided substantial impetus to assess sediment $C_{org}$ concentrations and burial rates in vegetated coastal ecosystems, which recently have been widely reviewed[9].

$C_{org}$ generally represents a minor fraction (2–3%) of buried material within mangrove and seagrass sediments[10,11] (although this is highly variable[12]), the rest being siliciclastic and carbonate particles. A global assessment of the concentration of inorganic carbon concluded that $C_{inorg}$ can exceed $C_{org}$ concentration in seagrass sediments[13]. Seagrass and mangrove plants do not calcify per se; however, they provide habitats for an abundant associated calcifying fauna and flora (e.g., crabs, sea stars, snails, bivalves, calcified algae, foraminifera), whose shells and skeletons may be deposited and buried in the sediment along with the plant litter, and the organic and inorganic particles imported from adjacent ecosystems.

Counterintuitively, $CaCO_3$ production represents a source of $CO_2$ to the atmosphere, as calcification produces $CO_2$ with a ratio of ~0.6 mol of $CO_2$ emitted per mol of $CaCO_3$ precipitated[14]. This has led to the argument that high $CaCO_3$ burial may partially offset $CO_2$ sequestration associated with $C_{org}$ burial in some seagrass meadows and mangrove forests[15]. However, there are several caveats that affect these arguments and render inferences on the role of Blue Carbon ecosystems as net $CO_2$ sinks or sources inconclusive[13,16], based on the comparison of $C_{org}$ and $C_{inorg}$ sediment burial rates. To date, very few articles report the burial rates of $CaCO_3$ in mangrove and seagrass ecosystems[15–17], and the role of $CaCO_3$ burial in sediments and $CO_2$ emissions depends on the balance between dissolution and production. If $CaCO_3$ dissolution equals local calcification, then the burial of $CaCO_3$ is supported exclusively by allochthonous inputs and is neutral in terms of $CO_2$ emissions or sequestration. If dissolution exceeds local calcification, then $CaCO_3$ dynamics add to the $CO_2$ sink capacity of Blue Carbon ecosystems, even if $CaCO_3$, which must be subsidized from allochthonous sources, is buried in the sediments. Only if $CaCO_3$ dissolution is lower than local calcification does $CaCO_3$ burial result in $CO_2$ emissions.

Here we address the current gap in global estimates of $C_{inorg}$ burial in seagrass and mangrove ecosystems by providing first estimates of contemporary (last century) $C_{inorg}$ burial rates. We rely on a compilation and analysis of data on sediment chronologies (i.e., including radiometric dating of sediment cores with $^{210}Pb$) and $C_{inorg}$ concentrations from around the world (Fig. 1). We compare burial, calcification and dissolution rates in three locations where most of the carbon mass balance terms were available. We then address the role of $CaCO_3$ burial in $CO_2$ emissions by resolving the source of the $CaCO_3$ buried in seagrass meadows as either allochthonous or autochthonous (i.e., from associated flora and fauna). We conclude that the high amounts of $CaCO_3$ found in Blue Carbon sediments can not be converted into $CO_2$ emissions.

## Results

**Global disparities in Blue Carbon sediments.** $CaCO_3$ supports an important part of sediment accretion rates (SARs) in seagrass ecosystems, although with large geographical disparities and a non-normal distribution (Shapiro–Wilks test, $p < 0.001$). Indeed, in 40% of global locations, the $CaCO_3$ concentration was under 10% dry weight (DW), whereas in 28% of locations the $CaCO_3$ content exceeded 80 %DW (see Supplementary Figure 1a). Overall, the median (interquartile range: IQR) global concentration of $CaCO_3$ in seagrass meadow sediments was 61 (56) %DW (mean ± SE of 54 ± 7).

In mangrove forests, we observe a large difference between the mean (± SE) and the median (IQR) $CaCO_3$ concentration: 21 ± 11% and 3 (31)%, respectively. This is explained by the strong non-normal distribution between the eight study locations examined, including a group of five locations with < 5 %DW $CaCO_3$ in their sediments and three locations with $CaCO_3$ contents between 20 and 75 %DW (Shapiro–Wilks test, $p < 0.001$, see Supplementary Fig. 1b). Converted into $C_{inorg}$ concentrations (after normalization for the sediment bulk density), we obtain median (IQR) $C_{inorg}$ concentrations in seagrass and mangrove sediments of 59 (66) and 1 (21) mg$C_{inorg}$ cm$^{-3}$, respectively (means ± SE of 63 ± 11 and 35 ± 17 mg$C_{inorg}$ cm$^{-3}$) (Fig. 2a).

Using the median SARs in seagrass and mangrove ecosystems compiled in this study (0.22 and 0.23 cm yr$^{-1}$, respectively; Fig. 2b), we estimate median (IQR) $C_{inorg}$ burial rates in seagrass and mangrove ecosystems of 87 (154) and 6 (207) g$C_{inorg}$ m$^{-2}$ yr$^{-1}$, respectively (means ± SE of 182 ± 94 and 90 ± 43 g$C_{inorg}$ m$^{-2}$ yr$^{-1}$) (Fig. 2c, Fig. 3). These values correspond to vertical accretion rates of $CaCO_3$ of the order of 0.1 and 0.001 cm yr$^{-1}$ in seagrass and mangrove ecosystems, respectively. Our mean SAR values agree with previously reported global values[1,3]. However, our new estimates of burial rates are lower than the previous, indirect median estimate of $C_{inorg}$ burial rate of 108 g$C_{inorg}$ m$^{-2}$ yr$^{-1}$ (mean ± SE of 126 ± 31 g$C_{inorg}$ m$^{-2}$ yr$^{-1}$) reported by Mazarrasa et al.[13].

**Global annual burial rates of $C_{inorg}$.** Scaling up to the global seagrass coverage (150,000–600,000 km$^2$)[9], the annual burial rate of $C_{inorg}$ ranged from 13 to 52 Tg$C_{inorg}$ yr$^{-1}$ for the twentieth century (Table 1). Partitioning between tropical and non-tropical seagrass meadows as in Mazarrasa et al.[13] showed that 90% of the global $C_{inorg}$ burial occurs in the tropics (Table 1). In seagrass meadows, our estimates of global burial of $C_{inorg}$ are equivalent to 31–55% of the available estimates of contemporary $C_{org}$ burial rates (48–112 Tg$C_{org}$ yr$^{-1}$)[1,7], depending on the estimated global seagrass coverage considered. If all buried $CaCO_3$ is locally produced (i.e., of autochthonous origin), the burial rates of $C_{inorg}$ in seagrass meadows would represent emissions of 8–37 TgC yr$^{-1}$ to the atmosphere and thus would offset their role as $CO_2$ sinks through the sequestration of $C_{org}$ by ~17–33%.

The extent of global mangrove coverage yields a median burial rate of 0.8 Tg$C_{inorg}$ yr$^{-1}$ (mean of 13 Tg$C_{inorg}$ yr$^{-1}$) (Table 1). This value should be considered as a first-order estimate because of the scarcity of data available on $C_{inorg}$ burial rates in mangroves and because of the non-normal distribution between $CaCO_3$-rich and $CaCO_3$-poor mangrove sediments (Supplementary Figure 1b). When comparing with the global $C_{org}$ burial rates estimate of 31 Tg$C_{org}$ yr$^{-1}$ [3], the median $C_{inorg}$ burial rates would correspond to a negligible reduction of net atmospheric $CO_2$ uptake. However, assuming that all sedimentary $CaCO_3$ was produced in situ, the $C_{inorg}$ burial rates can largely outweigh $C_{org}$ burial in $CaCO_3$-rich mangroves. For example, the $C_{inorg}$ burial rate corresponds to 10–20 times the $C_{org}$ burial rate in the Arabian Peninsula[17–19].

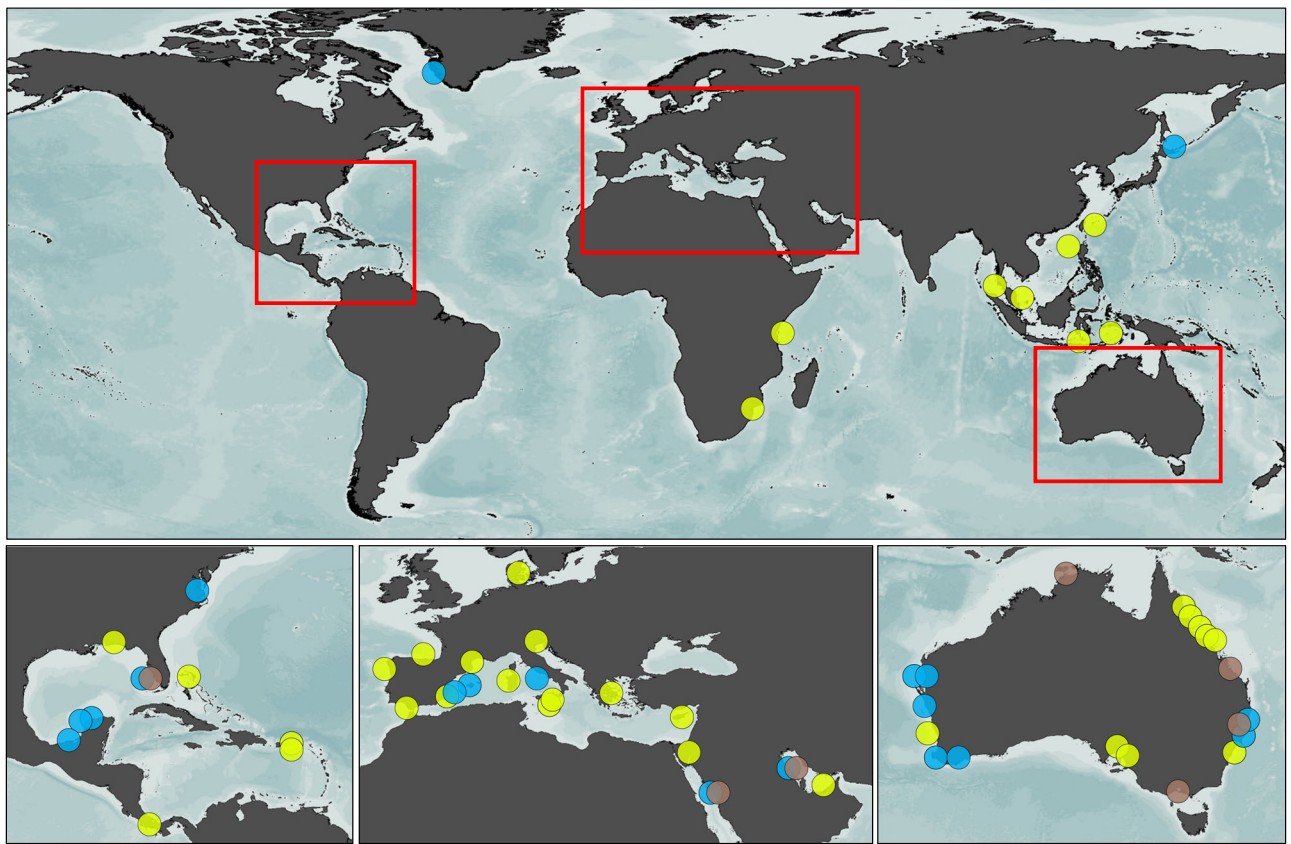

**Fig. 1** World map of sediment cores locations. Brown circles: mangrove cores locations; blue: seagrass cores locations; yellow: seagrass cores non-dated but with inorganic carbon content measured[13]

## Discussion

$CaCO_3$ burial in Blue Carbon ecosystems is the balance between inputs (autochthonous and allochthonous) and losses (dissolution and export). Assessments of the mass balance of $CaCO_3$ in seagrass meadows are few and none have been reported, to our knowledge, in mangrove forests. For seagrass ecosystems, we assessed the balance between calcification, dissolution and burial of $CaCO_3$ in three locations: the Balearic Islands, Spain[20,21], Shark Bay in Western Australia[22] and Florida Bay, USA[23,24] (Table 2).

The most comprehensive assessment of seagrass carbon budgets is that reported for a Mediterranean *Posidonia oceanica* meadow at Magalluf (Mallorca Island, Spain)[20,21,25,26]. In this meadow, Barrón et al.[21] estimated a net $CO_2$ uptake by primary production of 8.4 gC $m^{-2}$ $yr^{-1}$. This estimate was corroborated by the $C_{org}$ burial rate, estimated independently, at $9 \pm 2$ $gC_{org}$ $m^{-2}$ $yr^{-1}$ [27]. Barrón et al.[21] also estimated net calcification rates of 51 $gCaCO_3$ $m^{-2}$ $yr^{-1}$, corresponding to 6 $gC_{inorg}$ $m^{-2}$ $yr^{-1}$. This amount of calcification would result in a $CO_2$ emission of 3.6 gC $m^{-2}$ $yr^{-1}$ (0.6-fold the net calcification[14]). The $CO_2$ emission by calcification therefore represents an offset of 40% of the $CO_2$ uptake from net primary production (thereby yielding a total $CO_2$ sequestration of 4.8 gC $m^{-2}$ $yr^{-1}$ [21]). However, the $C_{inorg}$ burial rate in this meadow is estimated here at 226 $gC_{inorg}$ $m^{-2}$ $yr^{-1}$. This is two orders of magnitude greater than the net calcification rate of 6 $gC_{inorg}$ $m^{-2}$ $yr^{-1}$ [21](Table 2). This implies that about 90% of the $CaCO_3$ burial in this seagrass meadow must be supported by allochthonous inputs. Therefore, calculation of the $CO_2$ sequestration by comparing $C_{org}$ and $C_{inorg}$ burial rates or stocks would have concluded that this meadow is a strong source of $CO_2$, whereas estimates of calcification rates and net primary production concludes that it is a sink (as confirmed independently through air–sea flux measurements[26]).

Similarly, in Shark Bay, the burial of $C_{inorg}$ is four times higher than the independently reported net calcification rate[22] (Table 2). This again could require large allochthonous carbonate inputs.

In Florida Bay, the low ratio between $C_{org}$ and $C_{inorg}$ concentration in the sediment (g $cm^{-3}$) implied that seagrass meadows may be a net source of $CO_2$ to the atmosphere[15]. However, such assessment requires consideration of the full carbon mass balance, including accounting for allochthonous inorganic carbon inputs and the balance between calcification and dissolution in the meadows. The contemporary $C_{inorg}$ burial rates in Florida Bay are approximately ninefold higher than the global median, whereas median SAR is about fourfold higher than estimated globally, in an area where 80% of the sediment dry mass is composed of $CaCO_3$. However, attempts to assess the gross or net calcification rates in the area yielded values one and two orders of magnitude lower than the estimated $CaCO_3$ burial rates (Table 2)[23,24]. In contrast, past geological work in the Bay has suggested that it is a net producer of $CaCO_3$[28]. It is likely to be that some areas within this large Bay act as sources of $CaCO_3$ and some others as sinks, helping explain the discrepancy between reported production and burial estimates. Hence, internal redistribution of $CaCO_3$ production within Florida Bay needs to be considered when drawing inferences on the role of seagrass meadows from sediment composition.

These three example locations are in areas close to coral reefs and/or terrestrial lithogenic sources of $CaCO_3$. We could not find estimates of calcification rates (net or gross) in areas without external sources of $CaCO_3$. The discrepancies between calcification rates and burial rates in the three example locations could indicate that an important fraction of $CaCO_3$ burial (> 90%) is supported by $CaCO_3$ produced elsewhere and trapped in the seagrass sediments. This conclusion is consistent with

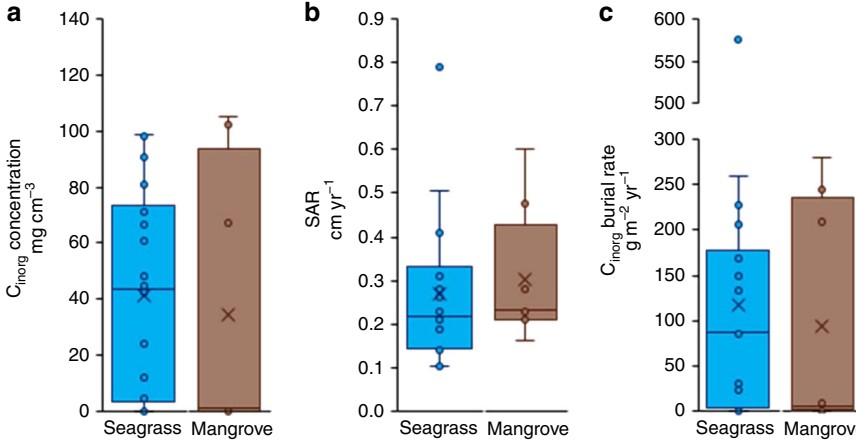

**Fig. 2** Sediment cores data. **a** Inorganic carbon ($C_{inorg}$) concentration, **b** sediment accumulation rates (SAR), and **c** $C_{inorg}$ burial rates. The x represents the mean. Bars are the first and last quartile

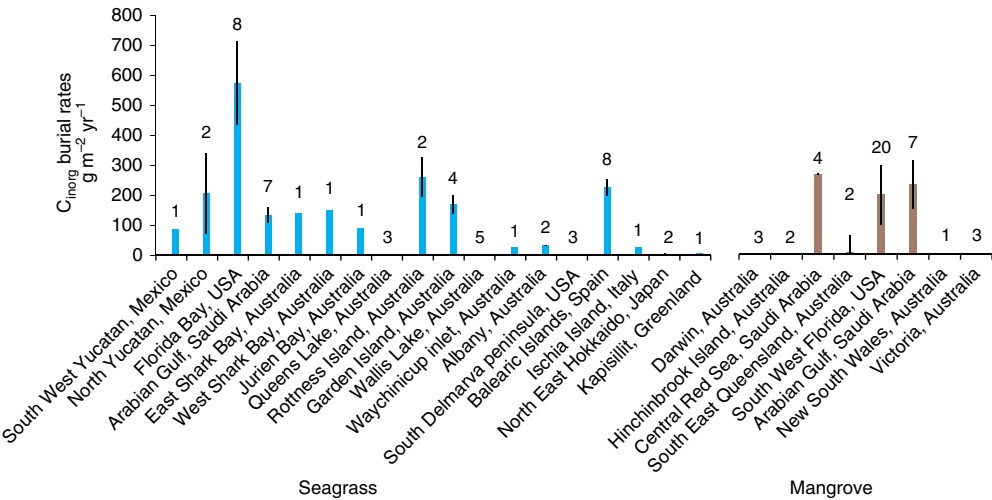

**Fig. 3** Inorganic carbon burial rates in all locations. Mean $C_{inorg}$ burial rates in all locations in sediment cores for seagrass meadows and mangrove forests, organized from low to high latitudes. Bars are the SE. Labels are the number of cores per location

| Table 1 Median (mean) global $C_{inorg}$ burial rates for seagrass meadows and mangrove forests considering one, and, for seagrass, two world regions (tropical and higher latitudes) | | | | | |
|---|---|---|---|---|---|
| | | **Burial rate, (TgC$_{inorg}$ yr$^{-1}$)** | | | |
| | | **Global** | **Tropical** | **Higher lat.** | **Sum** |
| Seagrass | This study | 13(27)–52(109) | 14(41)–57(163) | 1(3)–5(14) | 15(43)–62(177) |
| | Mazarassa et al.[13] | 19(28)–65(79) | | | |
| Mangrove | This study | 0.8(12) | | | |

comparable $C_{inorg}$ concentrations within and outside seagrass meadows, whereas, in contrast, $C_{org}$ concentrations are higher in seagrass sediments[13]. A large role of $C_{inorg}$ import is also consistent with the large $CaCO_3$ export from coral reefs to adjacent waters, equivalent to 25–50% of the $CaCO_3$ production, predominantly to reef lagoons[27], where seagrass meadows and mangroves often grow. Mangroves, seagrass and saltmarsh ecosystems are likely to be sites of net carbonate dissolution. Roots of marine plants release organic compounds and oxygenate the sediments during the day, promoting microbial aerobic remineralization of organic matter, thereby increasing sedimentary respiratory $CO_2$[29,30] and/or stimulating the re-oxidation of reduced metabolites. These processes result in the release of strong acids (e.g., $H_2SO_4$, $HNO_3$)[31–33], which leads to $CaCO_3$ dissolution in the sediment[34,35] (although re-precipitation can occur[34]).

**Table 2 Burial rates of CaCO₃ compared to calcification rates in seagrass ecosystems**

| | Community production rate of CaCO₃ | | Community net calcification rate | | Sediment | | |
|---|---|---|---|---|---|---|---|
| | $gCaCO_3\ m^{-2}\ yr^{-1}$ | $gC_{inorg}\ m^{-2}\ yr^{-1}$ | $gCaCO_3\ m^{-2}\ yr^{-1}$ | $gC_{inorg}\ m^{-2}\ yr^{-1}$ | %CaCO₃ | $gCaCO_3\ m^{-2}\ yr^{-1}$ | $gC_{inorg}\ m^{-2}\ yr^{-1}$ |
| Florida Bay, USA | 626[23,24] | 75 | 18[25] | 2.2 | 82 ± 2 | 4792 ± 756 | 756 ± 91 |
| Balearic Islands, Spain | 68[20] | 8 | 51[21] | 6 | 81 ± 3 | 1886 ± 214 | 226 ± 30 |
| West Shark Bay, Australia | 375 ± 62[22] | 45 ± 7 | 295[22] | 35 | 60 ± 5 | 1240 ± 232 | 149 ± 30 |

Comparison between seagrass-associated community production rate of carbonate (obtained from standing stock assessments and leaves or calcifiers turnover rates) and community net calcification rates (balance between calcification and dissolution, calculated from variations of total alkalinity) from the literature, and carbonate burial rate in three locations with carbonate-rich sediments

Dissolution of $CaCO_3$ might also be influenced by the $CO_2$ system in the water column of Blue Carbon ecosystems. Respiration and photosynthesis of the flora and fauna, together with sediment redox processes in seagrass and mangrove ecosystems, strongly influence the chemistry of the water column, generating large diel amplitudes of the saturation state for $CaCO_3$ ($\Omega$) with a tendency towards dissolution or the reduction of calcification at nighttime, amplified at low tide[36–40]. The dissolution of allochthonous $CaCO_3$ in carbonate platform areas, caused directly or indirectly by metabolism of the marine vegetation and associated biota, leads to a reduction in $pCO_2$ through the release of fossilized total alkalinity. This sink of atmospheric $CO_2$ should be incorporated into the Blue Carbon framework. A recent assessment considers alkalinity addition through the dissolution of allochthonous carbonate as a very effective geoengineering approach to remove atmospheric $CO_2$ and mitigate climate change[41,42].

Similarly, saltmarshes are not known to host high levels of calcifying organisms but can accumulate $CaCO_3$ from allochthonous sources. In arid tropical saltmarshes of the Western Arabian Gulf, dominated by succulent shrubs, a concentration of $CaCO_3$ of 57 ± 8% in sediments and a contemporary burial rate of $C_{inorg}$ of $100 ± 15\ gC_{inorg}\ m^{-2}\ yr^{-1}$ (mean ± SE) were found[17]. In a temperate saltmarsh of the Western Scheldt estuary in the Netherlands, a concentration of $CaCO_3$ of 14 ± 1% and a high contemporary burial rate of $467 ± 99\ gC_{inorg}\ m^{-2}\ yr^{-1}$, mostly due to high SAR ($1.1 ± 0.3\ cm\ yr^{-1}$), were measured[43]. Yet, this does not imply that $CaCO_3$ dynamics have a negligible role in saltmarsh carbon budgets, as they may still act as sites of net dissolution of $CaCO_3$, adding to the removal of $CO_2$ associated with $C_{org}$ burial. A dissolution rate of $24–96\ gC_{inorg}\ m^{-2}\ yr^{-1}$ was estimated in the sediment of the saltmarshes of the Eastern Scheldt estuary, corresponding to ~85% of the $C_{inorg}$ burial rate[44].

To further examine the conclusion that Blue Carbon ecosystems are sites of substantial allochthonous $CaCO_3$ deposition, based on existing mass balances for Blue Carbon sediments, we examined (qualitatively) the relationship between the $CaCO_3$ % DW in sediments and the presence/absence of sources of $CaCO_3$ adjacent to the coring locations, including coral reefs and terrestrial lithogenic sources of $CaCO_3$ (Fig. 4, see dataset in Supplementary Information). Seagrass and mangrove ecosystems without potentially large adjacent allochthonous $CaCO_3$ sources have a remarkably lower median (IQR) sediment $CaCO_3$ content of 4 (15) and 1 (1) %DW (means ± SE of 11 ± 4 and 1.7 ± 0.8 % DW), respectively, compared with 59 (51) and 61 (27) %DW (means ± SE of 56 ± 5 and 53 ± 16 %DW) when at least one allochthonous $CaCO_3$ source was present (Fig. 3). For sediments in seagrass meadows, the presence of coral reefs (*t*-value = 4.68, df = 48.5, $p < 0.0001$) and lithogenic sources (*t*-value = 4.76, df = 57.3, $p < 0.0001$) increased the amount of $CaCO_3$ in the sediment. However, there was a significant interaction between these factors (*t*-value = − 3.29, df = 53.2, $p = 0.0018$), because the $CaCO_3$ %

DW in the presence of both allochthonous sources was less than would be expected if these variables were additive. The presence/absence of coral reefs and lithogenic sources accounted for 36% of the variation in $CaCO_3$, whereas the random variables (study, lithology grouping and marine province) accounted for 54% of the variation in $CaCO_3$ (see Methods for model description). Mangrove sediment samples showed a similar pattern to the seagrass meadows and the presence of allochthonous sources had a marginally significant positive effect on the amount of $CaCO_3$ in the sediment (*t*-value = 4.29, df = 1.81, $p = 0.0596$). The presence/absence of a $CaCO_3$ source accounted for 71% of the variation of in $CaCO_3$ within mangrove sediments, whereas the random variables accounted for 20% of the variation in $CaCO_3$. In testing for biases of outlying cores and studies, we found that one study from Western Florida had an outlying data point that disproportionality skewed the results. The study from Western Florida had relatively low $CaCO_3$ but did have an allochthonous source of $CaCO_3$. When this study was removed from the analysis, the presence of an allochthonous source became significant (*t*-value = 7.92, df = 4.16, $p = 0.0012$). This highlights the need for more studies in mangrove sediments to determine the global influence of allochthonous sources on $CaCO_3$ content.

In seagrass meadows, the median (IQR) $C_{inorg}$ burial rate found in areas where no allochthonous sources were identified was 1 (13) $gC_{inorg}\ m^{-2}\ yr^{-1}$ (mean ± SE of 8 ± 4), only 1.1% of the global median. This contrast is consistent with our hypothesis that much of the $C_{inorg}$ buried in seagrass and mangrove sediments is allochthonous. It explains the non-normal distribution of $CaCO_3$ concentrations observed in sediments of seagrass and mangroves (Supplementary Figure 1), and indicates that the import of $CaCO_3$ from carbonate-forming ecosystems or adjacent karstic areas is the norm[27]. The global burial rate of $C_{inorg}$ in seagrass meadows is between a third to a half of their $C_{org}$ burial rate[1,7], whereas for mangroves our first estimate of global $C_{inorg}$ burial rates is only 3% of the $C_{org}$ burial rate[3]. If the buried $CaCO_3$ and $C_{org}$ in seagrass sediments were produced entirely in situ, $C_{inorg}$ burial would offset up to a third of $CO_2$ sequestration through $C_{org}$ burial, particularly in tropical seagrass ecosystems where ~90% of the global $C_{inorg}$ burial occurs. However, imbalances between production, dissolution and burial, and the observation of much higher $CaCO_3$ concentrations in sediments near lithogenic formations and coral reefs, suggest that, where present, allochthonous $CaCO_3$ inputs are substantial and support most of the net $CaCO_3$ burial.

Locally, despite supporting significant $CaCO_3$ burial, Blue Carbon ecosystems may be sites where imported $CaCO_3$ dissolves, strengthening rather than weakening the capacity of these ecosystems to sequester $CO_2$. Whereas there is emphasis on apportioning the sources of autochthonous and allochthonous $C_{org}$ in Blue Carbon sediments (up to 50% of the buried $C_{org}$)[4,9], determining the sources of $CaCO_3$ in Blue Carbon sediments is just as important, to resolve the role of vegetated coastal

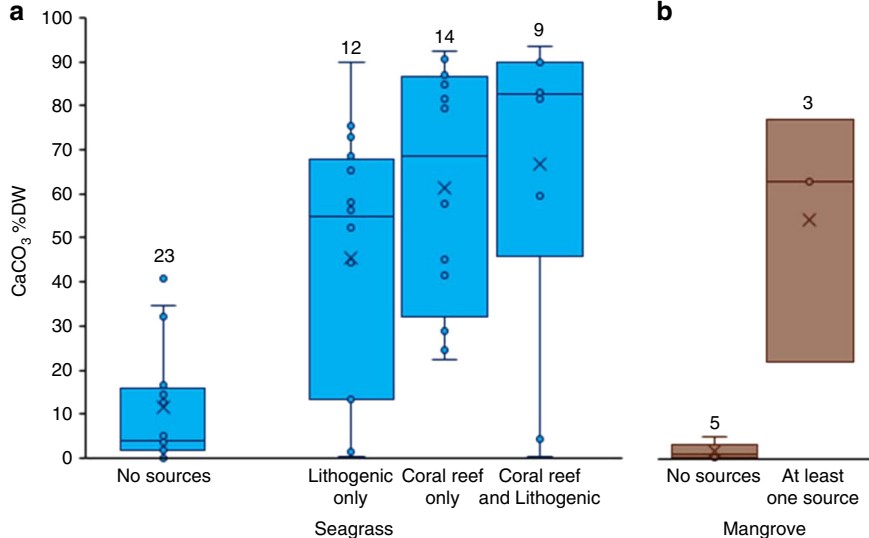

**Fig. 4** Allochthonous sources and inorganic carbon in sediments. **a** %DW of carbonates ($CaCO_3$) in seagrass sediments depending on the presence of potential allochthonous sources (lithogenic and/or coral reefs). All data distributions for locations with allochthonous sources are significantly different to the distributions for locations without allochthonous sources (Mann–Whitney $U$-tests, all $p < 0.001$). **b** %DW $CaCO_3$ in mangrove sediments with and without allochthonous sources (coral reef and lithogenic source). Number on top of the box plots indicate the number of locations. The $x$ represents the mean. Bars are the first and last quartile

ecosystems as $CO_2$ sinks and, hence, their potential to support climate change mitigation. The current focus on $C_{org}$ budgets in vegetated coastal ecosystems needs to be augmented with integrative assessments of organic and inorganic carbon fluxes and budgets, including both allochthonous and autochthonous inputs. Moreover, these assessments must consider the sources and fate of carbon exchanged between Blue Carbon and adjacent ecosystems, as Blue Carbon ecosystems export important amounts of their organic production[45–47] but also import significant amounts of $CaCO_3$ and organic matter from adjacent sources. A comparison of paired vegetated and unvegetated sediment $CaCO_3$ % DW showed that vegetated and adjacent unvegetated sediments have similar carbonate concentrations, both using standard parametric statistics (general linear model (GLM), $t$-value = 1.32, df = 83.1, $p = 0.191$) and meta-analysis ($z$-value = 0.88, $p = 0.379$; Supplementary Fig. 2A,B), which also showed no evidence for reporting bias (all points within the 95% confidence lines of the funnel plot, Supplementary Fig. 2C. This provides further support to the hypothesis that much of the carbonate buried in vegetated coastal sediments derives from allochthonous sources rather than being produced within the habitat.

Inorganic carbon burial in Blue Carbon ecosystems has been overlooked, with the rates compiled here representing the first direct estimates reported in the literature. These estimates confirm that seagrass ecosystems, and to lesser extent mangrove ecosystems, are intense sites of $CaCO_3$ burial, supporting sediment accretion. $CaCO_3$ burial is a fundamental process supporting the role of Blue Carbon ecosystems in climate change adaptation, which is underpinned by their capacity to rapidly accrete sediments, reducing relative SLR by raising the seafloor[1,17].

## Methods

**Calculation of the $C_{inorg}$ accretion rate**. We searched the peer-reviewed literature for data on sediment cores dated with $^{210}Pb$, including $CaCO_3$ or $C_{inorg}$ concentration in seagrass and mangrove sediments. Search terms on Google Scholar were seagrass OR mangrove AND 210Pb OR SAR OR sediment accretion rate. We then searched returned articles that contained data on SAR and $CaCO_3$ or $C_{inorg}$ data. We found only one study presenting $CaCO_3$ content in a dated sediment core. However, we found 15 and 22 studies with SAR for seagrass and mangrove

sediments, respectively. To obtain the $CaCO_3$ or $C_{inorg}$ concentrations needed to calculate $C_{inorg}$ burial rates, we used the database of Mazarrasa et al. [13], which was the most recent exhaustive compilation of sediment cores from Blue Carbon habitats, for data on $CaCO_3$ in seagrass sediments. We also contacted experts in Blue Carbon studies (published studies using cores from Blue Carbon habitats) for unpublished $CaCO_3$ sediment concentration data (see data and references in Supplementary Data 1). In total, we compiled 42 and 53 $^{210}Pb$ dated cores with $CaCO_3$ content in mangrove and seagrass ecosystems, respectively (see PRISMA checklist and flow diagram[48] in Supplementary Note 1).

The SARs (cm $yr^{-1}$) from the literature were re-calculated according to the constant flux–constant sedimentation model[49], to have a coherent and comparable dating system between all cores. The $CaCO_3$ concentration (% sediment DW) was calculated as the mean between all slices younger than 1900, for cores with the contemporary $^{210}Pb$ chronologies. The $C_{inorg}$ concentration in sediment (g$C_{inorg}$ $m^{-3}$) was calculated from the dry bulk density (g $m^{-3}$) and the percentage of $CaCO_3$ content (using sediment DW), considering a mass ratio of 12% carbon in $CaCO_3$. The $C_{inorg}$ burial rate (g$C_{inorg}$ $m^{-2}$ $yr^{-1}$) was then calculated as the product of the SAR and the $C_{inorg}$ concentration for each sediment core. Cores with negligible content of $CaCO_3$ were also included in the calculation (see Supplementary Figure 1).

All cores from the same site or area and with similar presence or absence of allochthonous sources of $CaCO_3$ (see below) were treated as replicates for a global location and averaged for the analysis (geologic grouping). For seagrass, the 51 cores dated with $^{210}Pb$ were grouped into 17 locations (Figs. 2, 3). For mangroves, we compiled a total of 42 cores dated with $^{210}Pb$ in 8 locations (Figs. 2, 3). Seagrass locations ranged from tropical to sub-arctic locations, with 50% of estimates derived from tropical and subtropical locations and 50% from higher latitudes. Mangrove sediment derived mostly from subtropical locations (seven out of eight locations), particularly in Australia and the Arabian Peninsula (Supplementary Figure 2).

**Determination of the influence of allochthonous sources of $CaCO_3$**. We analysed the influence of the presence/absence of proximity of coral reefs and continental surface lithology (qualitative data), as potential allochthonous sources of $CaCO_3$ in seagrass and mangrove sediments (in %DW) (see dataset in Supplementary Data 1). For seagrass, we expanded our dataset by including $CaCO_3$ concentrations from 264 cores compiled by Mazarrasa et al.[13], reaching a total of 341 cores with measured $CaCO_3$ %DW.

We estimated the presence/absence of coral reefs using the map of the global distribution of warm-water coral reefs compiled by UNEP-WCMC[50] and the presence/absence of nearby lithogenic sources using the global lithology map of Hartmann and Moosdorf[51] and the world soil map of the FAO/UNESCO (http://www.fao.org/soils-portal/en/). The coring locations were associated with climate regions following the Köppen–Geiger classification system[52].

**Statistical analysis**. All data distributions were tested for normality to determine the most reliable central tendency measured with Shapiro–Wilks normality test (Statistica, Dell Software). None of the datasets of SAR, $C_{inorg}$ concentration, $C_{inorg}$

burial rate or $CaCO_3$ %DW were normally distributed (all $p < 0.05$). We therefore chose to use the median (IQR) as the most appropriate description of central tendency. Traditional meta-analysis tools, which calculate effect sizes to standardize the difference between control and experimental treatments, thereby allowing comparison among disparate response variables and weighting to account for unequal variance among studies, could not be used for this analysis for multiple reasons. These reasons include that the question posed and the studies available did not include experimental designs with paired control and experimental plots required for effect size calculations, that there was a single response variable facilitating direct comparison and data integration, and, most importantly, that we used the raw data for each core. Instead, we ran a statistical test using a mixed effect GLM to determine the effect of coral reefs and lithogenic sources on the $CaCO_3$ %DW of the sediment. For sediments within seagrass meadows, the GLM included two fixed factors (presence/absence of coral reefs and of lithogenic sources), as well as the interaction between the two factors. For sediment within mangrove forests, the GLM included one fixed factor (presence/absence of allochthonous sources), because replication did not exist for all combinations of the two factors. The data had unequal samples among studies and studies were not evenly distributed around the globe (Fig. 1), which could result in pseudo-replication and biased results. To account for the data structure and minimize non-independence, we included three separate random variables, which included study, lithology grouping and marine province. The marine province was determined for each sample location using the marine provinces of the world as defined by Spalding et al.[53]. Separate models where run for seagrass and mangrove sites. The statistical model was produced using the lmer function within the lme4 package[54] and $p$-values were calculated with the lmerTest package[55]. The $R^2$ was calculated for the fixed and random effects using the r.squared GLMM function in the MuMIn package[56]. The response variable was log transformed, which improved the model fit compared with raw data. The model fit was assessed by plotting the $Q$–$Q$ plot (linear relationship) and the fitted values compared with the residuals (random distribution). To test whether individual cores or studies were biased and having a disproportionate influence on findings, we systematically removed any studies that contained outlying samples as determined from being outside the 95% confidence interval for the fitted values vs. residuals comparison using the plot model function from sjPlot package[57]. This analysis was conducted in R version 3.4.2.

Reporting bias and its effect on findings is an important consideration for meta-analyses[58] and when the result from a meta-analysis is not the same as it would have been if data from all correctly conducted studies were included in the analysis[59]. A main cause of reporting bias is not publishing research because of a lack of merit as determined by the researcher, reviewer or editor[60]. As indicated by the data inclusion flow diagram (Supplementary Fig. 3), researchers often measured but did not publish data on soil $CaCO_3$ content and authors needed to be directly contacted for these results. In addition, the researchers not only provided information from published studies but also unpublished data on $CaCO_3$ content (10 of 51 seagrass studies included in the analysis were not published). For these reasons, it is unlikely to be that our findings were affected by reporting bias. A subset of data collected for this study included the appropriate information to run both a GLM and a traditional meta-analysis (effect size could be calculated between paired data). The data included information from nine studies that measured the $CaCO_3$ content of sediment from both vegetated and unvegetated habitats. There were 92 core samples with 32 from unvegetated and 60 from vegetated habitats (Supplementary Fig. 2A). The GLM followed the same procedures as detailed in the main text, except it had only two random factors, study and marine province, because study and lithology grouping differed in only one instance. For the meta-analysis, the data were paired for each study and the mean $CaCO_3$ %DW, number of samples and SD were calculated for vegetated and unvegetated cores for each study. Two studies included in the GLM were removed for the meta-analysis, because they only had one core for an unvegetated habitat and SD could not be calculated, leaving seven comparisons for this analysis. Hedges' $g$ was calculated for the effect size following Borenstein et al.[60] (Eqs. 1–3) and a variance for each effect size was also calculated ($V_g$)[59] (Eq. 4), as indicated by equations:

$$\text{Hedges}' g = \frac{(X_E - X_C)J}{\text{SD}_{\text{pooled}}} \quad (1)$$

$$\text{SD}_{\text{pooled}} = \sqrt{\frac{(n_E - 1)(\text{SD}_E)^2 + (n_C - 1)(\text{SD}_C)^2}{n_E + n_C - 2}} \quad (2)$$

$$J = 1 - \frac{3}{4(n_E + n_C - 2) - 1} \quad (3)$$

$$V_g = \frac{n_E + n_C}{n_E \times n_C} + \frac{g^2}{2(n_E + n_C)} \quad (4)$$

$X_E$ and $X_C$ are the mean ($n$ is sample size) of vegetated and unvegetated sediments, along with $\text{SD}_{\text{pooled}}$ and $J$, which accounts for biases associated with different sample sizes. The meta-analysis included the same two random variables as the GLM and was conducted using the rma.mv function from the metafor package[61].

**Calculation of global yearly burial rates of $C_{\text{inorg}}$.** The global annual burial of inorganic carbon ($TgC_{\text{inorg}} \text{ yr}^{-1}$) in seagrass meadows was calculated as the product of the global median $C_{\text{inorg}}$ burial rates and the estimated global seagrass area, which ranges from 150,000 to 600,000 $km^2$[29]. We also calculated the global annual burial of $C_{\text{inorg}}$ as the sum of separate calculations for tropical and arid climates and meadows at higher latitude climates. Median $C_{\text{inorg}}$ burial rates were calculated for tropical (core locations with tropical and hot desert climates) and non-tropical areas (temperate, continental and polar climates) and multiplied by the global seagrass cover range under the assumption that 2/3 of the seagrass area is in the tropical and subtropical zone[13]. The global annual burial of inorganic carbon ($TgC_{\text{inorg}} \text{ yr}^{-1}$) in mangroves was calculated as the product of the global median $C_{\text{inorg}}$ burial rates and the estimated global mangrove cover of 137,760 $km^2$[62].

## Data availability
The dataset is available as Supplementary Data 1.

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

## Acknowledgements

This research was supported by King Abdullah University of Science and Technology (KAUST) through baseline funding and workshop funding to C.M.D. Support from the Australian Research Council through grants LIEF Project LE170100219, DE160100443, DE170101524, DP150103286, DP150102092, DP160100248, DE130101084, LP160100242 and LE140100083 is acknowledged. J.J.M. was supported by the Netherlands Earth System Science Center. J.W.F. was supported by the US National Science Foundation through the Florida Coastal Everglades Long-Term Ecological Research programme under Grant Number DEB-1237517. D.K.-J. received financial support from the Independent Research Fund Denmark (8021-00222B, CARMA) and the COCOA project under the BONUS programme, funded by the EU 7th Framework Programme and the Danish Research Council. A.A.-O. was supported by an "Obra Social la Caixa" fellowship (LCF/BQ/ES14/10320004). A.A.-O. and P.M. acknowledge the support by the Generalitat de Catalunya (grant 2017 SGR-1588). This work is contributing to the ICTA 'Unit of Excellence' (MinECo, MDM2015-0552). H.K.'s input is a contribution to the CESEA project (NE/L001535/1) funded by NERC. T.K., K.W. and T.T. were supported by JSPS KAKENHI (18H04156) and the Environment Research and Technology Development Fund (S-14) of the Ministry of the Environment, Japan. J.M.S. was supported by the National Science Foundation under South Florida Water, Sustainability and Climate grant EAR-1204079. I.M. was supported by a Juan de la Cierva Formación post-doctoral fellowship from the Spanish Ministry of Science, Innovation and Universities.

## Author contributions

C.M.D. and V.S. conceived and designed this work. V.S. wrote the manuscript with support from N.R.G., P.I.M., D.T.M., J.J.M., O.S. and C.M.D. V.S., N.R.G., P.I.M., D.T.M., J.J.M., O.S., H.A., A.A.-O., M.C., B.D.E., J.W.F., H.K., D.K.-J., T.K., P.S.L., C.E.L., N.M., P.M., M.A.M., I.M., K.J.M.G., M.P.J.O., C.J.S., I.R.S., J.M.S., T.T., K.W. and C.M.D. contributed data and quality check was done by A.A.-O. and P.M. Analysis of data was performed by V.S. and N.R.G. V.S., N.R.G., P.I.M., D.T.M., J.J.M., O.S., H.A., A.A.-O., M.C., B.D.E., J.W.F., H.K., D.K.-J., T.K., P.S.L., C.E.L., N.M., P.M., M.A.M., I.M., K.J.M.G., M.P.J.O., C.J.S., I.R.S., J.M.S., T.T., K.W. and C.M.D. discussed and commented on the manuscript.

## Additional information

**Competing interests:** The authors declare no competing interests.

