## [Peer Review File · Nature Communications]

Reviewers' comments:

Reviewer #1 (Remarks to the Author):

This is an interesting and large scale synthesis but substantial reworking is required to meet the criteria for robust meta-analysis (Cf Handbook of meta-analysis in ecology and evolution; Gurevitch et al, 2018 Nature).

In terms of synthesis, the extended data of Mazarrasa et al. 2015 contains the appropriate information for the derivation and synthesis of effect sizes- either standardised mean difference or mean difference. These effects should have been calculated and combined using random effects meta-analysis, accompanied by forest and funnel plots. Meta-regressions and model selection procedures should have been used to explore heterogeneity in effects.

Critical in this approach is the use of weighting which is absent from the authors ad-hoc approach to synthesis.

Details of the acquisition of information are also sparse and a source of major concern. The databases searched, search strings, inclusion criteria and inter-rater reliability must all be explicit, especially in the absence of any consideration of publication bias.

The authors have gone to great lengths to assemble an important data set. I would strongly urge them to engage with the systematic review and meta-analysis community to ensure that biases are minimised in synthesis of this information.

Reviewer #2 (Remarks to the Author):

The manuscript submitted by V. Saderne entitled "Role of Carbonate Burial in "Blue Carbonate" Budgets" attempts to address mismatches in local carbonate production and measurements of net carbonate burial in productive coastal ecosystems. Their results suggest that much of the carbonate buried in seagrass and mangrove systems is imported, and this allochthonous carbonate should be recognized as external when accounting for the sequestration capacity of the local ecosystem.

This distinction is a valid one, and is important for understanding the productivity and sequestration potential of these ecosystems as a whole. The study represents a step forward in understanding the mass balance of carbon in and out of these coastal ecosystems. I think this article is worthy of publishing, although I have a few main points that I would like clarification on.

First, the authors argue for significant allochthonous input of CaCO_3 only when there are significant sources of carbonate material nearby. Their mass balances are conducted in three systems where there is a large input of external carbonate. It would help to see a mass balance conducted at a site where local dynamics drive the whole system, e.g. a “No Sources” site from Figure 4. This would provide a counter-example to their dataset, and potentially constrain the role of dissolution in these systems.

Secondly, I have an issue with the framing of carbonate dissolution in the context of these systems. There is an estimate from shark bay that a 21% dissolution rate could indeed be supported by allochthonous CaCO_3 import, but it also must be supported by significant aerobic respiration in the sediments. The total amount of carbon sequestered therefore must still be driven by the net amount of organic carbon buried. It is unclear to me if bicarbonate produced by the neutralization of respired CO_2 due to carbonate dissolution should be included in the mass balance framework of this study.

The role of dissolution is only hypothesized in mangrove systems, where there is little to no carbonate production data. The assertion of mangroves being a source of DIC is true only for an oxic system; if the system goes anoxic in the sediments, then anoxic respiration of CO_2 would produce DIC:Alkalinity in at least a 1:1 ratio and would be indistinguishable from respiration-driven carbonate dissolution. It would help to justify the assertion that these systems are completely oxic throughout the sediment column.

Finally, there is a significant difference between contemporary- and long-term carbonate burial rates. Do the authors have a hypothesis for why burial rates are so different between these two timescales? Is there any difference in the burial rate of organic carbon as well? Can these longer-term budgets help reconcile mismatches in contemporary budgets and rates, or is there still a significant mismatch between carbonate production and burial?

Some smaller points:

Line 33 - awkward phrasing, improper "if-then" statement.

Lines 42-43: Mixed tense. Stick with present tense as in the rest of the paragraph, or switch all to past tense

Line 45 - Need to define autochthonous

Line 48: supports

Line 84: Do you mean bimodal? Binomial is more like a normal distribution...

Line 91: The introduction of the three sites are out of order with respect to their presentation in the following text.

Line 105: Evidence that Corg production is all local helps the argument here. Is this true at all sites? How would import and/or export of locally produced Corg bias this study's findings with respect to the role of relative Corg vs. Cinorg burial?

Line 124: "areas close to coral reefs"

Lines 130-133: Awkward. Please rephrase.

Line 183: This last line feels like a non-sequitur. I understand why it is mentioned, but needs more justification and/or introduction.

Figure 2: What is the difference between Corg concentration between ^{14}C and ^{210}Pb data? In general in these box plots, what is the x, is this the mean?

Table 2, caption and numbers: There are no negative values that I can see, even taking into account errors.

Supplementary Figure 1: I am not sure that there is sufficient data density to call Figure S1b a "bimodal distribution". At the very least it is non-normal, or perhaps you see two peaks in CaCO_3 concentration?

Reviewer #1 (Remarks to the Author):

This is an interesting and large-scale synthesis, but substantial reworking is required to meet the criteria for robust meta-analysis (Cf Handbook of meta-analysis in ecology and evolution; Gurevitch et al, 2018 Nature).

In terms of synthesis, the extended data of Mazarrasa et al. 2015 contains the appropriate information for the derivation and synthesis of effect sizes- either standardised mean difference or mean difference. These effects should have been calculated and combined using random effects meta-analysis, accompanied by forest and funnel plots. Meta-regressions and model selection procedures should have been used to explore heterogeneity in effects.

Critical in this approach is the use of weighting which is absent from the authors ad-hoc approach to synthesis.

Details of the acquisition of information are also sparse and a source of major concern. The databases searched, search strings, inclusion criteria and inter-rater reliability must all be explicit, especially in the absence of any consideration of publication bias.

The authors have gone to great lengths to assemble an important data set. I would strongly urge them to engage with the systematic review and meta-analysis community to ensure that biases are minimised in synthesis of this information.

We appreciate the suggestions made by the reviewer. In response to the reviewer's comments, we have incorporated many changes to the analysis which enhanced both the clarity of the methods and robustness of the findings. We have now included both requirements as stated in PRISMA (Preferred Reporting Items for Systematic Reviews and Meta-Analyses¹²; <http://www.prisma-statement.org/>), which are suggested in Gurevitch et al. (2018)⁶³. This included answering a checklist of 27 items (in supplement) and presenting a diagram of the information flow and an explicit statement of how the data was gathered and included in the analysis (in supplement), as well as including all publications that were a source of analysed data in the main text.

Following the reviewer suggestions, we clarified the search method in the material and methods (in addition to adding the flow diagram):

New Ms Ln 214-225 “We searched the peer-reviewed literature for data on sediment cores dated with ¹⁴C and/or ²¹⁰Pb, including CaCO₃ or C_{inorg} concentration in seagrass and mangrove sediments. Search terms on Google Scholar were “seagrass” OR “mangrove” AND “²¹⁰Pb” OR “¹⁴C” OR “SAR” OR “sediment accretion rate”. We then searched returned articles that contained data on SAR and CaCO₃ or C_{inorg} data. We found only one study presenting CaCO₃ content in a dated sediment core. However, we found 15 and 22 studies with sediment accretion rates (SAR) for seagrass and mangrove sediments, respectively. To obtain the CaCO₃ or C_{inorg} concentrations needed to calculate C_{inorg} burial rates, we used the database of Mazarrasa et al. (2015)¹³, which was the most recent exhaustive compilation of sediment cores from blue carbon habitats, for data on CaCO₃ in seagrass sediments. We also contacted experts in blue carbon studies (published studies using cores from blue carbon habitats) for unpublished CaCO₃ sediment concentration data (see data and references in supplementary dataset). In total, we compiled 42 and 53 ²¹⁰Pb dated cores with CaCO₃ content in mangrove and seagrass ecosystems, respectively; and 71 ¹⁴C dated cores with CaCO₃ content in seagrass ecosystems only (see PRISMA checklist and flow diagram⁵² in supplementary material).”

Including these materials greatly enhanced the transparency and reproducibility of the study, which are both necessary for proper systematic reviews.

The dataset in this study is actually more detailed and focused than most metanalyses in that it includes raw data for each core and we focus on only a single response variable, calcium carbonate content of sediment. Analysing the raw data with the same response variable allowed us to run a general linear model, which was more appropriate than a meta-analysis statistical approach. Meta-analysis was

developed to analyse data when only a mean and variance (used to weight the means as mentioned by the reviewer) are available from a study and when combining data with disparate response variables. We were very careful to incorporate all important aspects of the meta-analysis, including discussion of data availability and including random variables in the statistical model to minimize potential sources of non-independence among samples (in methods on lines 272 - 283). Specifically, we repeated the analysis with raw data using a mixed effects general linear model. The model included two fixed factors (presence of coral reefs and calcium carbonate bedrock) and 3 random variables to incorporate potential non-independence of multiple cores sampled within the same study, the same marine province, and the same geologic region. Although this is not exactly the analysis suggested by the reviewer (effect sizes, either standardized mean difference or mean difference), we think the reviewer will agree that our analysis is more appropriate given that we have raw data with the same response variable and that paired sampling did not exist (sites with and without coral or with and without bedrock were not sampled within studies, but were analyzed here *post hoc*, but as a paired set). Finally, following the reviewer's suggestion to meet the criteria to produce a robust meta-analysis, we also included discussion of publication bias (new Ms ln. 199 - 205 and 260 - 299). Although we could not run traditional tests for publication bias (funnel plots) because we used raw data, the usual mechanisms for publication bias (Koricheva et al. 2013, Chapter 14)⁶⁴ did not exist for this study given that authors had to be contacted for data on calcium carbonate content (data was not included in publication of core data) and these authors were asked for and provided data on studies that were unpublished (approximately 20% of the seagrass studies in the analysis were not published). However, we did test whether specific studies were disproportionately influencing results by systematically removing any studies with outliers from the analysis (specified on New Ms ln. 172 - 176).

Although we could not run a traditional meta-analysis on the data comparing source presence and C_{inorg} in sediment, we did analyse an additional data set comprised of data collected by this study to demonstrate that C_{inorg} was not influenced by the presence of biogenic habitats, and we analysed these data with both a GLM and a traditional meta-analysis, as suggested by the reviewer. The data for this new analysis included paired samplings of vegetated and unvegetated cores, which rendered this dataset suitable for a traditional meta-analysis using effect size as well as its associated variance. We included all elements that the reviewer suggested, including effect size calculations, a funnel plot and forest plot (see Added in new MS: 199 – 205; ln. 257 – 309 and in supplementary material). We think these results are not only interesting and relevant to the study, but also demonstrate the utility, complementarity and robustness of both analyses (GLM and effect-size based).

In summary, the reviewer's comments greatly improved the manuscript, which now clearly documents all aspects normally covered in a meta-analysis. However, as explicitly stated earlier, our data set differed from the usual ones available for meta-analysis (e.g. we had access to and used raw data rather than descriptive statistics such as median or mean, and our analysis did not include, except for the data set discussed above, paired data allowing effect size calculations).

Note also that we have incorporated a new co-author, Dr. Nathan Geraldi, an experienced meta-analysis ecologist, for his contribution to the new analyses conducted.

The new text reporting these analyses (GLM and meta-analyses) as well as details on data search and sources are described below:

Lines 163 – 178: “). For sediments in seagrass meadows, the presence of coral reefs (t-value = 4.68, df = 48.5, $p < 0.0001$) and lithogenic sources (t-value = 4.76, df = 57.3, $p < 0.0001$) increased the amount of $CaCO_3$ in the sediment. There was a significant interaction between these factors (t-value = -3.29, df = 53.2, $p = 0.0018$) because the $CaCO_3$ %DW in the presence of both allochthonous sources was less than would be expected if these variables were additive. The statistical model explained 90% of the variation in $CaCO_3$ %DW and the fixed factors accounted for 36% of the variation, while the random variables accounted for 54%. Mangrove sediment samples showed a similar pattern to the seagrass meadows, and the presence of allochthonous sources had a marginally significant, positive effect on the amount of $CaCO_3$ in the sediment (t-value = 4.29, df = 1.81, $p = 0.0596$). The statistical model explained 91% of the variation in sediment $CaCO_3$ and the fixed factors accounted for 71% of the variation, while the random variables accounted for 20% of the variation. In testing for biases of outlying cores

and studies, we found that one study from Western Florida had an outlying data point that disproportionality skewed the results. The study from Western Florida had relatively low CaCO₃ but did have an allochthonous source of CaCO₃. When this study was removed from the analysis, the presence of an allochthonous source became significant (t-value = 7.92, df = 4.16, $p = 0.0012$). This highlights the need for more studies in mangrove sediments to determine the global influence of allochthonous sources on CaCO₃ content.“

Added in new MS: 199 – 205: “A comparison of paired vegetated and unvegetated sediment CaCO₃ %DW showed that vegetated and adjacent unvegetated sediments have similar carbonate concentrations, both using standard parametric statistics (GLM, t-value = 1.32, df = 83.1, $p = 0.191$) and meta-analysis (z-value = 0.88, $p = 0.379$; Supplement Fig. 2. A. B.), which also showed no evidence for reporting bias (all points within the 95% confidence lines of the funnel plot, Supplement Fig. 2. C). This provides further support to the hypothesis that much of the carbonate buried in vegetated coastal sediments derives from allochthonous sources rather than being produced within the habitat.”

Old MS ln. 183 – 189:” We searched the peer-reviewed literature for data on sediment cores dated with ¹⁴C and/or ²¹⁰Pb, including CaCO₃ or C_{inorg} concentration in seagrass and mangrove sediments. Search terms on Google Scholar were “seagrass” OR “mangrove” AND “²¹⁰Pb” OR “¹⁴C” OR “SAR” OR “sediment accretion rate”. We then searched returned articles that contained data on SAR and CaCO₃ or C_{inorg} data. We found only one study presenting CaCO₃ content in a dated sediment core. However, we found 15 and 22 studies with sediment accretion rates (SAR) for seagrass and mangrove sediments, respectively. To obtain the CaCO₃ or C_{inorg} concentrations needed to calculate C_{inorg} burial rates, we used the database of Mazarrasa et al. (2015)¹³, which was the most recent exhaustive compilation of sediment cores from blue carbon habitats, for data on CaCO₃ in seagrass sediments and contacted experts in blue carbon studies (published studies using cores from blue carbon habitats) for unpublished CaCO₃ sediment concentration data (see data and references in supplementary dataset).”

Was changed to:

New MS ln. 214 – 225: “We searched the peer-reviewed literature for data on sediment cores dated with ¹⁴C and/or ²¹⁰Pb, including CaCO₃ or C_{inorg} concentration in seagrass and mangrove sediments. Search terms on Google Scholar were “seagrass” OR “mangrove” AND “²¹⁰Pb” OR “¹⁴C” OR “SAR” OR “sediment accretion rate”. We then searched returned articles that contained data on SAR and CaCO₃ or C_{inorg} data. We found only one study presenting CaCO₃ content in a dated sediment core. However, we found 15 and 22 studies with sediment accretion rates (SAR) for seagrass and mangrove sediments, respectively. To obtain the CaCO₃ or C_{inorg} concentrations needed to calculate C_{inorg} burial rates, we used the database of Mazarrasa et al. (2015)¹³, which was the most recent exhaustive compilation of sediment cores from blue carbon habitats, for data on CaCO₃ in seagrass sediments. We also contacted experts in blue carbon studies (published studies using cores from blue carbon habitats) for unpublished CaCO₃ sediment concentration data (see data and references in supplementary dataset). In total, we compiled 42 and 53 ²¹⁰Pb dated cores with CaCO₃ content in mangrove and seagrass ecosystems, respectively; and 71 ¹⁴C dated cores with CaCO₃ content in seagrass ecosystems only (see PRISMA checklist and flow diagram⁵² in supplementary material). “

Added in new Ms ln. 257 – 309: “All data distributions were tested for normality to determine the most reliable central tendency measured with Shapiro-Wilks normality test (Statistica, Dell Software). None of the datasets of SAR, C_{inorg} concentration, C_{inorg} burial rate or CaCO₃ %DW were normally distributed (all $p < 0.05$). We therefore chose to use the median (IQR) as the most appropriate description of central tendency. Traditional meta-analysis tools, which calculate effect sizes to standardize the difference between control and experimental treatments, thereby allowing comparison among disparate response variables and weighting to account for unequal variance among studies, could not be used for this analysis for multiple reasons. These reasons include that the question posed and the studies available did not include experimental designs with paired control and experimental plots required for effect size calculations, that there was a single response variable facilitating direct comparison and data integration and, most importantly, that we used the raw data for each core. Instead, we ran a statistical test using a mixed effect general linear model (GLM) to determine the effect of coral reefs and lithogenic sources on

the CaCO₃ %DW of the sediment. For sediments within seagrass meadows, the GLM included two fixed factors (presence / absence of coral reefs and of lithogenic sources), as well as the interaction between the two factors. For sediment within mangrove forests, the GLM included one fixed factor (presence / absence of allochthonous sources) because replication did not exist for all combinations of the two factors. The data had unequal samples among studies and studies were not evenly distributed around the globe (Fig. 1) which could result in pseudo-replication and biased results. To account for the data structure and minimize non-independence, we included 3 separate random variables, which included study, lithology grouping, and marine province. The marine province was determined for each sample location using the marine provinces of the world as defined by Spalding et al. (2007)⁵⁸. Separate models were run for seagrass and mangrove sites. The statistical model was produced using the *lmer* function within the *lme4* package⁵⁹ and p-values were calculated with the *lmerTest* package⁶⁰. The R² was calculated for the fixed and random effects using the *r.squaredGLMM* function in the *MuMIn* package⁶¹. The response variable was log transformed, which improved the model fit compared with raw data. The model fit was assessed by plotting the Q-Q plot (linear relationship) and the fitted values compared to the residuals (random distribution). To test if individual cores or studies were biased and having a disproportionate influence on findings we systematically removed any studies that contained outlying samples as determined from being outside the 95% confidence interval for the fitted values vs. residuals comparison using the *plotModel* function from *sjPlot* package⁶². This analysis was conducted in R version 3.4.2.

Reporting bias and its effect on findings is an important consideration for meta-analyses⁶³ and when the result from a meta-analysis is not the same as it would have been if data from all correctly conducted studies were included in the analysis⁶⁴. A main cause of reporting bias is not publishing research because of a lack of merit as determined by the researcher, reviewer or editor⁶⁴. As indicated by the data inclusion flow diagram (supplementary Fig. 3), researchers often measured but did not publish data on soil CaCO₃ content and authors needed to be directly contacted for these results. In addition, the researchers not only provided information from published studies but also unpublished data on CaCO₃ content (10 of 51 seagrass studies included in the analysis were not published). For these reasons, it is unlikely that our findings were affected by reporting bias. A subset of data collected for this study included the appropriate information to run both a GLM and a traditional meta-analysis (effect size could be calculated between paired data). The data included information from nine studies that measured the CaCO₃ content of sediment from both vegetated and unvegetated habitats. There were 92 core samples with 32 from unvegetated and 60 from vegetated habitats (Supplementary Fig. 2. A.). The GLM followed the same procedures as detailed in the main text except it had only two random factors, study and marine province, because study and lithology grouping differed in only one instance. For the meta-analysis, the data was paired for each study and the mean CaCO₃ %DW, number of samples, and standard deviation was calculated for vegetated and unvegetated cores for each study. Two studies included in the GLM were removed for the meta-analysis because they only had one core for an unvegetated habitat and standard deviation could not be calculated, leaving seven comparisons for this analysis. Hedges' *g* was calculated for the effect size following Borenstein et al. (2009)⁶⁵ and a variance for each effect size was also calculated (*V_g*)⁶⁴, as indicated by equations:

$$\text{Hedges}'g = \frac{(X_E - X_C)J}{SD_{pooled}}$$

$$SD_{pooled} = \sqrt{\frac{(n_E - 1)(SD_E)^2 + (n_C - 1)(SD_C)^2}{n_E + n_C - 2}}$$

$$J = 1 - \frac{3}{4(n_E + n_C - 2) - 1}$$

$$Vg = \frac{n_E + n_C}{n_E \times n_C} + \frac{g^2}{2(n_E + n_C)}$$

X_E and *X_C* are the mean (*n* is sample size) of vegetated and unvegetated sediments, along with *SD_{pooled}* and *J* which accounts for biases associated with different sample sizes. The meta-analysis included the same two random variables as the GLM and was conducted using the *rma.mv* function from the *metafor* package⁶⁶.”

Reviewer #2 (Remarks to the Author):

The manuscript submitted by V. Saderne entitled “Role of Carbonate Burial in “Blue Carbon” Budgets” attempts to address mismatches in local carbonate production and measurements of net carbonate burial in productive coastal ecosystems. Their results suggest that much of the carbonate buried in seagrass and mangrove systems is imported, and this allochthonous carbonate should be recognized as external when accounting for the sequestration capacity of the local ecosystem.

This distinction is a valid one, and is important for understanding the productivity and sequestration potential of these ecosystems as a whole. The study represents a step forward in understanding the mass balance of carbon in and out of these coastal ecosystems. I think this article is worthy of publishing, although I have a few main points that I would like clarification on.

First, the authors argue for significant allochthonous input of CaCO_3 only when there are significant sources of carbonate material nearby. Their mass balances are conducted in three systems where there is a large input of external carbonate. It would help to see a mass balance conducted at a site where local dynamics drive the whole system, e.g. a “No Sources” site from Figure 4. This would provide a counter-example to their dataset, and potentially constrain the role of dissolution in these systems.

We wish to warmly thank Dr. Subhas for his review and highlighting some unclear points of our MS that needed to be addressed.

Indeed, it would be great to have a mass balance in a “no-source” site, for which we also have an associated burial rate. Unfortunately, there are few locations where burial rates of CaCO_3 have been measured and few locations where net or gross calcification rates have been assessed, so the combination of both requirements is extremely rare. We could not find a single site without CaCO_3 external sources where calcification rates and burial rates of CaCO_3 are documented (as, logically, scientists have yet to document these where carbonates are important).

We added new MS ln 125: “We could not find estimates of calcification rates (net or gross) in any area without external sources of CaCO_3 . “

Secondly, I have an issue with the framing of carbonate dissolution in the context of these systems. There is an estimate from shark bay that a 21% dissolution rate could indeed be supported by allochthonous CaCO_3 import, but it also must be supported by significant aerobic respiration in the sediments. The total amount of carbon sequestered therefore must still be driven by the net amount of organic carbon buried. It is unclear to me if bicarbonate produced by the neutralization of respired CO_2 due to carbonate dissolution should be included in the mass balance framework of this study.

We agree that we may have lacked clarity in our original MS on these points. We agree that the atmospheric CO_2 uptake linked to the generation of TA coming from CaCO_3 dissolution should indeed be included in the CO_2 balance of the ecosystems (and the text has been modified to articulate this, new Ms ln 132-146 see below). Dissolution is related to aerobic mineralisation (and re-oxidation of reduced metabolites originating from anaerobic mineralization) and it is possible that the dissolution rate would be related to the OM burial rate, because the latter scales with OM mineralisation. However, in the absence of data, this remains speculative and we would rather refrain from assuming this to be the case without clear supporting evidence.

The role of dissolution is only hypothesized in mangrove systems, where there is little to no carbonate production data. The assertion of mangroves being a source of DIC is true only for an oxic system; if the system goes anoxic in the sediments, then anoxic respiration of CO_2 would produce DIC: Alkalinity in at least a 1:1 ratio and would be indistinguishable from respiration-driven carbonate dissolution. It would help to justify the assertion that these systems are completely oxic throughout the sediment column.

We agree with these comments and realize that we didn't express ourselves clearly in the MS. What we meant is that the $p\text{CO}_2$ in mangrove swamp water is generally well above 400 μatm , particularly at night, and in interaction with the tides, as seen in many studies such as in e.g. Sippo et al., 2016⁴⁰. This is also true in seagrass meadows (Challener et al., 2016)⁴¹. Therefore, these habitats are very prone to night-time dissolution of superficial CaCO_3 . A different process happens in the sediment, and as you rightfully commented, it depends on the oxygenation of the sediment. In vegetated systems with radial oxygen loss from roots (particularly root tips), oxic decomposition of organic matter leads to CO_2 generation and therefore acidification of the porewater, which may lead to dissolution of carbonates, particularly during daytime, when photosynthesis occurs (opposite to what could be hypothesised for the water column). Note that, within the framework of this article, we do not discuss all the processes susceptible to generate or uptake total alkalinity in seagrass and mangrove sediments. Although that is a very exciting and important research question (see Sippo et al., 2016)⁴⁰, we have restricted the focus of this study to the CaCO_3 dissolution / calcification balance and the CO_2 uptake / release generated through TA emission / release.

Considering the two previous comments, we rewrote the whole paragraph:

new Ms In 132-146: "Indeed, mangroves, seagrass and saltmarsh ecosystems are likely to be sites of net carbonate dissolution. Roots of marine plants release organic compounds and oxygenate the sediments during the day, promoting microbial aerobic remineralization of organic matter, thereby increasing sedimentary respiratory CO_2 ³²⁻³⁴ and / or stimulating the re-oxidation of reduced metabolites. These processes result in the release of strong acids (e.g. H_2SO_4 , HNO_3)³⁵⁻³⁷ that leads to CaCO_3 dissolution in the sediment^{38,39}, (although re-precipitation can occur³⁸).

Dissolution of CaCO_3 might also be influenced by the CO_2 system in the water column of Blue Carbon ecosystems. Respiration and photosynthesis of the flora and fauna, together with sediment redox processes in seagrass and mangrove ecosystems, strongly influence the chemistry of the water column, generating large diel amplitudes of the saturation state for CaCO_3 (Ω) with a tendency toward dissolution or the reduction of calcification at night-time, amplified at low tide^{34,40-43}. The dissolution of allochthonous CaCO_3 in carbonate platform areas, caused directly or indirectly by metabolism of the marine vegetation and associated biota, leads to a reduction in $p\text{CO}_2$ through the release of "fossilised" total alkalinity. This sink of atmospheric CO_2 should be incorporated into the blue carbon framework. A recent assessment considers alkalinity addition through the dissolution of allochthonous carbonate as a very effective geo-engineering approach to remove atmospheric CO_2 and mitigate climate change^{44,45}."

Finally, there is a significant difference between contemporary- and long-term carbonate burial rates. Do the authors have a hypothesis for why burial rates are so different between these two timescales? Is there any difference in the burial rate of organic carbon as well? Can these longer-term budgets help reconcile mismatches in contemporary budgets and rates, or is there still a significant mismatch between carbonate production and burial?

Indeed, and as stated in the MS (In. 71-73), lower long-term (^{14}C) SAR are commonly observed. There are no doubts within the Blue Carbon community of the credibility of the modern and long term SAR obtained by ^{210}Pb and ^{14}C dating technique. There are hypothesis indeed explaining these differences, the main being that the SAR are increasing with rates of sea level rise and that, globally, blue carbon ecosystems are keeping pace with the modern acceleration of sea level rise. See Saderne et al., 2018³⁴ as an example of a recent study discussing that aspect and the role of carbonate burial in supporting SAR. Indeed, according to the IPCC report, the rates of sea level rise have doubled from about 0.17 cm yr^{-1} for the period 1901 – 2010 to 0.32 cm yr^{-1} for the period 1993 – 2010. In contrast, SLR for the preindustrial period are estimated to be 0.01 cm yr^{-1} . These considerations are beyond the scope of this article, but we believe it is indeed important that, as we stated in the MS, the SAR means / medians observed in our collection of cores match the published global estimates.

There are differences between short term and long-term burial rates of C_{org} reported in the literature, such as in Serrano et al., 2016 for example. This is due to lower long-term SAR, but also to lower concentrations of C_{org} in deep sediments partially due to the slow remineralisation of organic matter over time.

We do not have estimates of calcification rates in seagrass meadows for pre-industrial times, so we cannot compare long-term burial with ancient calcification rates in any of the locations for which we have ^{14}C dated cores. If we compare the long-term burial rates of C_{inorg} with contemporary $CaCO_3$ production rates for the three locations for which we have both dating methods, Florida Bay, West Shark Bay and the Balears Islands, $CaCO_3$ production would explain 73%, 475% and 38% of the $CaCO_3$ burial. However, comparing ancient burial rates with contemporary calcification rates doesn't change the conclusion that modern burial rates of $CaCO_3$ cannot be converted to CO_2 emissions, in the context of climate change that is of interest to us.

Serrano, Oscar, Aurora M. Ricart, Paul S. Lavery, Miguel-Angel Mateo, Ariane Arias-Ortiz, Mohammad Rozaimi, Andy DL Steven, and Carlos Duarte. "Key biogeochemical factors affecting soil carbon storage in Posidonia meadows." (2016).

Some smaller points:

Line 33 - awkward phrasing, improper "if-then" statement.

Changed in new MS Ln 33-35: "To date, very few articles report the burial rates of $CaCO_3$ in mangrove and seagrass ecosystems^{15,16,17}, and the role of $CaCO_3$ burial in sediments and CO_2 emissions depends on the balance between dissolution and production."

Lines 42-43: Mixed tense. Stick with present tense as in the rest of the paragraph, or switch all to past tense.

All verbs were changed to the present tense.

Line 45 - Need to define autochthonous

New MS Ln 44 – 46: "We then address the role of $CaCO_3$ burial in CO_2 emissions by resolving the source of the $CaCO_3$ buried in seagrass meadows as either allochthonous or autochthonous (i.e. from associated flora and fauna)."

Line 48: supports

We corrected it in the New MS.

Line 84: Do you mean bimodal? Binomial is more like a normal distribution...

Indeed, we changed binomial for non-normal.

Line 91: The introduction of the three sites are out of order with respect to their presentation in the following text.

We rectified the order of each site introduction.

Line 105: Evidence that C_{org} production is all local helps the argument here. Is this true at all sites? How would import and/or export of locally produced C_{org} bias this study's findings with respect to the role of relative C_{org} vs. C_{inorg} burial?

The partitioning of buried C_{org} in Blue Carbon ecosystems between local vs external C sources is an important question that is not within the scope of this article, as assessing autochthonous vs. allochthonous carbon, often done through stable isotope mixing models (e.g. Kennedy et al. 2010)⁴⁸, has its own limitations which are now being debated in search of an improved approach. The answer is that not all buried C_{org} is originating from the seagrass and mangroves habitats in general, but Kennedy et al., 2010 estimated it to be around 50% globally. In the context of Ln105, considering that part of the

C_{org} buried was produced outside the seagrass beds would only enlarge the difference between C_{org} and C_{inorg} burial.

Ln. 191 – 199: “Whereas there is emphasis on apportioning the sources of autochthonous and allochthonous C_{org} in Blue Carbon sediments (up to 50% of the buried C_{org})^{9,48}, determining the sources of CaCO_3 in Blue Carbon sediments is just as important in order to resolve the role of vegetated coastal ecosystems as CO_2 sinks and, hence, their potential to support climate change mitigation. The current focus on C_{org} budgets in vegetated coastal ecosystems needs to be augmented with integrative assessments of organic and inorganic carbon fluxes and budgets, including both allochthonous and autochthonous inputs. Moreover, these assessments must consider the sources and fate of carbon exchanged between Blue Carbon and adjacent ecosystems, as Blue Carbon ecosystems export important amounts of their organic production^{49–51}, but also import significant amounts of CaCO_3 and organic matter from adjacent sources. “

Line 124: “areas close to coral reefs”

We added “to” to the sentence.

Lines 130-133: Awkward. Please rephrase.

The entire paragraph has been rewritten for clarity: new Ms Ln 132-146: “Indeed, mangroves, seagrass and saltmarsh ecosystems are likely to be sites of net carbonate dissolution. Roots of marine plants release organic compounds and oxygenate the sediments during the day, promoting microbial aerobic remineralization of organic matter, thereby increasing sedimentary respiratory CO_2 ^{32–34} and / or stimulating the re-oxidation of reduced metabolites. These processes result in the release of strong acids (e.g. H_2SO_4 , HNO_3)^{35–37} that leads to CaCO_3 dissolution in the sediment^{38,39}, (although re-precipitation can occur³⁸).

Dissolution of CaCO_3 might also be influenced by the CO_2 system in the water column of Blue Carbon ecosystems. Respiration and photosynthesis of the flora and fauna, together with sediment redox processes in seagrass and mangrove ecosystems, strongly influence the chemistry of the water column, generating large diel amplitudes of the saturation state for CaCO_3 (Ω) with a tendency toward dissolution or the reduction of calcification at night-time, amplified at low tide^{34,40–43}. The dissolution of allochthonous CaCO_3 in carbonate platform areas, caused directly or indirectly by metabolism of the marine vegetation and associated biota, leads to a reduction in pCO_2 through the release of “fossilised” total alkalinity. This sink of atmospheric CO_2 should be incorporated into the blue carbon framework. A recent assessment considers alkalinity addition through the dissolution of allochthonous carbonate as a very effective geo-engineering approach to remove atmospheric CO_2 and mitigate climate change^{44,45}. “

Line 183: This last line feels like a non-sequitur. I understand why it is mentioned, but needs more justification and/or introduction.

We removed the sentence in the new MS.

Figure 2: What is the difference between C_{org} concentration between ^{14}C and ^{210}Pb data? In general in these box plots, what is the x, is this the mean?

The difference arises from the fact that the ^{14}C and ^{210}Pb are measured in different cores taken in different parts of the world, therefore leading to differences in CaCO_3 concentrations. The x is the mean, which we added in the figure caption: “The x represents the mean.”

Table 2, caption and numbers: There are no negative values that I can see, even taking into account errors.

Indeed, that is an error from our side, we removed the mention of negative values in the table caption.

Supplementary Figure 1: I am not sure that there is sufficient data density to call Figure S1b a “bimodal distribution”. At the very least it is non-normal, or perhaps you see two peaks in CaCO_3 concentration?

We have changed “bimodal” for “non-normal” throughout the MS.

Reviewers' comments:

Reviewer #1 (Remarks to the Author):

I thank the authors for responding positively to my comments. The inclusion of the PRISMA statement and additional analyses have increased both the transparency and robustness of the conclusions.

Reviewer #2 (Remarks to the Author):

This review is for the revised manuscript entitled "Role of carbonate burial in "Blue Carbon" budgets. The authors have mostly satisfactorily responded to the first reviews, and I think the manuscript is very close to being acceptable for publication. There are a few minor issues that I am confident the authors can deal with that I enumerate below.

The main one has to do with ^{210}Pb vs ^{14}C mass accumulation rates. I acknowledge that such discrepancies have been observed before. However, if the ^{14}C MARs do not match with ^{210}Pb estimates, and the current study primarily addresses the relationship between ~ 100 year MARs and essentially instantaneous measurements of net ecosystem calcification, I encourage the authors to consider removing the ^{14}C -based MARs entirely from the manuscript. If the authors wish to address the mismatch directly, I think they might be a useful part of the discussion. Indeed, you have proposed a hypothesis for the offset in your response to my previous comments. But as it stands, the discrepancy does not add anything to the points the manuscript is trying to make, as acknowledged in the response to my previous comments.

I also find it interesting that, in your comments, you say that current calcification in Shark Bay is $\sim 475\%$ larger than the long-term mass accumulation rates. This suggests that, in the long term, Shark Bay may be a site of significant dissolution.

Some smaller points:

Line 1: Perhaps say Calcium carbonates

Line 7: should define Allocthonous

Line 30: "Large" feels awkward. Perhaps "large (or high) CaCO₃ burial rates" ?

Lines 43-46: "We compared... We then address" ... need to match tenses here.

Lines 71-72: Recall that to go from 90-80% CaCO₃, one needs to dissolve over half of the carbonate. Therefore, %CaCO₃ is a pretty poor measure of CaCO₃ preservation rate. This could play into the short- and long-term offsets in accumulation rate seen here.

Lines 95-97: Recommend not using passive voice. Perhaps, "We assessed the balance between calcification, dissolution, and burial of CaCO₃ in three seagrass ecosystem locations:"

Line 102: Awkward use of "CO₂ emissions from net calcification metabolism", especially because "net calcification metabolism" is then referred to later using the same value. Table 2 makes it clear that the actual calcification rate is different than the value given here, but this section could be made clearer.

Line 112: Where/how is this dissolution rate calculated? This sentence makes it sound like Table 2 shows dissolution rates when in fact this is something reported from a previous study.

Line 127: "Fraction of CaCO₃ burial"

Lines 136-137: choose either "," or "("

Lines 156-177: This section seems useful and is a good addition to the manuscript, but a reference should be made to the Materials and Methods section in terms of the statistical tests, and especially as to which variables are "fixed" versus "random" in the generalized linear model.

Table 1: A reference to the Methods section would be helpful for why the “Global” and “Sum” values are slightly off from one another.

Table 2: What is the difference between “Community production rate” and “Ecosystem net calcification rate”? These need to be defined in some way. For Florida Bay in particular, these two values are quite different from each other; for the others, they seem to be in better agreement.

Supplementary Figure 2: A) needs an axis label. All plots should be better defined in the figure caption, or the appropriate Methods section should be directly referenced for how the plots were constructed.

Reviewer #2 (Remarks to the Author):

This review is for the revised manuscript entitled “Role of carbonate burial in “Blue Carbon” budgets. The authors have mostly satisfactorily responded to the first reviews, and I think the manuscript is very close to being acceptable for publication. There are a few minor issues that I am confident the authors can deal with that I enumerate below.

The main one has to do with ^{210}Pb vs ^{14}C mass accumulation rates. I acknowledge that such discrepancies have been observed before. However, if the ^{14}C MARs do not match with ^{210}Pb estimates, and the current study primarily addresses the relationship between ~100-year MARs and essentially instantaneous measurements of net ecosystem calcification, I encourage the authors to consider removing the ^{14}C -based MARs entirely from the manuscript. If the authors wish to address the mismatch directly, I think they might be a useful part of the discussion. Indeed, you have proposed a hypothesis for the offset in your response to my previous comments. But as it stands, the discrepancy does not add anything to the points the manuscript is trying to make, as acknowledged in the response to my previous comments. I also find it interesting that, in your comments, you say that current calcification in Shark Bay is ~475% larger than the long-term mass accumulation rates. This suggests that, in the long term, Shark Bay may be a site of significant dissolution.

We wish to heartfully thank reviewer 2 for his comments on the manuscript. Indeed, the ^{14}C -based does not add anything to the point we are making in the MS and addressing the mismatch between ^{14}C and ^{210}Pb is not the scope of the article. A recent meta-analysis, Breithaupt et al., 2018 addressed the question of the discrepancies of the SAR between time scales, validating the observation of increasing SAR toward contemporary times. We followed the advice of reviewer 2 and removed all parts regarding the ^{14}C from the MS.

Breithaupt, J. L., Smoak, J. M., Byrne, R. H., Waters, M. N., Moyer, R. P., & Sanders, C. J. (2018). Avoiding timescale bias in assessments of coastal wetland vertical change. *Limnology and Oceanography*, 63(S1).

Old Ms ln 61 - 74.

“Using the median contemporary (last century – ^{210}Pb) SARs in seagrass and mangrove ecosystems compiled in this study (0.22 cm yr⁻¹ and 0.23 cm yr⁻¹, respectively; Fig. 2b), we estimate median (IQR) C_{inorg} burial rates in seagrass and mangrove ecosystems of 87 (154) and 6 (207) g C_{inorg} m⁻² yr⁻¹, respectively (means ± SE of 182 ± 94 and 90 ± 43 g C_{inorg} m⁻² yr⁻¹) (Fig. 2c, Fig. 3). These values correspond to vertical accretion rates of CaCO₃ of the order of 0.1 and 0.001 cm yr⁻¹ in seagrass and mangrove ecosystems, respectively.

Our dataset allowed us to also calculate the median long-term (^{14}C) burial rate of C_{inorg} in seagrass ecosystems, yielding 30 (84) g C_{inorg} m⁻² yr⁻¹ (mean ± SE of 62 ± 16 g C_{inorg} m⁻² yr⁻¹) (Fig. 2c), three-fold less than the contemporary C_{inorg} burial rates. Our new estimates of long-term and contemporary burial rates are lower than the previous, indirect median estimate of C_{inorg} burial rate of 108 g C_{inorg} m⁻² yr⁻¹ (mean ± SE of 126 ± 31 g C_{inorg} m⁻² yr⁻¹) reported by Mazarrasa et al. (2015)¹³. The difference between contemporary and long-term burial is driven by the three-fold difference in SARs between these timescales, as contemporary and older sediments did not differ significantly in C_{inorg} concentration (Mann-Whitney U test, p = 0.29; supplementary Table 1 for details). Such difference between long-term and contemporary SARs has previously been observed in coastal vegetated ecosystems¹⁷⁻¹⁹. Our mean SAR values are in agreement with previously reported global values^{1,3}.”

New Ms ln 60 - 66:

Using the median SARs in seagrass and mangrove ecosystems compiled in this study (0.22 cm yr⁻¹ and 0.23 cm yr⁻¹, respectively; Fig. 2b), we estimate median (IQR) C_{inorg} burial rates in seagrass and mangrove ecosystems of 87 (154) and 6 (207) g C_{inorg} m⁻² yr⁻¹, respectively (means ± SE of 182 ± 94 and 90 ± 43 g C_{inorg} m⁻² yr⁻¹) (Fig. 2c, Fig. 3). These values correspond to vertical accretion rates of CaCO₃

of the order of 0.1 and 0.001 cm yr⁻¹ in seagrass and mangrove ecosystems, respectively. Our mean SAR values agree with previously reported global values^{1,3}. However, our new estimates of burial rates are lower than the previous, indirect median estimate of C_{inorg} burial rate of 108 gC_{inorg} m⁻² yr⁻¹ (mean ± SE of 126 ± 31 gC_{inorg} m⁻² yr⁻¹) reported by Mazarrasa et al. (2015)¹³.”

Old Ms In 214 – 236

“We searched the peer-reviewed literature for data on sediment cores dated with ²¹⁰Pb, including CaCO₃ or C_{inorg} concentration in seagrass and mangrove sediments. Search terms on Google Scholar were “seagrass” OR “mangrove” AND “²¹⁰Pb” OR “SAR” OR “sediment accretion rate”. We then searched returned articles that contained data on SAR and CaCO₃ or C_{inorg} data. We found only one study presenting CaCO₃ content in a dated sediment core. However, we found 15 and 22 studies with sediment accretion rates (SAR) for seagrass and mangrove sediments, respectively. To obtain the CaCO₃ or C_{inorg} concentrations needed to calculate C_{inorg} burial rates, we used the database of Mazarrasa et al. (2015)¹³, which was the most recent exhaustive compilation of sediment cores from blue carbon habitats, for data on CaCO₃ in seagrass sediments. We also contacted experts in blue carbon studies (published studies using cores from blue carbon habitats) for unpublished CaCO₃ sediment concentration data (see data and references in supplementary dataset). In total, we compiled 42 and 53 ²¹⁰Pb dated cores with CaCO₃ content in mangrove and seagrass ecosystems, respectively (see PRISMA checklist and flow diagram⁵⁰ in supplementary material).

The SARs (cm yr⁻¹) from the literature were re-calculated according to the constant flux - constant sedimentation model⁵¹ in order to have a coherent and comparable dating system between all cores. The CaCO₃ concentration (% sediment dry weight) was calculated as the mean between all slices younger than 1900, for cores with the contemporary ²¹⁰Pb chronologies. The C_{inorg} concentration in sediment (gC_{inorg} m⁻³) was calculated from the dry bulk density (g m⁻³) and the percentage of CaCO₃ content (using sediment dry weight), considering a mass ratio of 12% carbon in CaCO₃. The C_{inorg} burial rate (gC_{inorg} m⁻² yr⁻¹) was then calculated as the product of the SAR and the C_{inorg} concentration for each sediment core. Cores with negligible content of CaCO₃ were also included in the calculation (see supplementary Fig. 1).

All cores from the same site or area and with similar presence or absence of allochthonous sources of CaCO₃ (see below) were treated as replicates for a global location and averaged for the analysis (geologic grouping). For seagrass, the 51 cores dated with ²¹⁰Pb were grouped into 17 locations (Fig. 2, 3). For mangroves, we compiled a total of 42 cores dated with ²¹⁰Pb in 8 locations (Fig. 2, 3). Seagrass locations ranged from tropical to sub-arctic locations, with 50% of estimates derived from tropical and subtropical locations and 50% from higher latitudes. Mangrove sediment derived mostly from subtropical locations (7 out of 8 locations), particularly in Australia and the Arabian Peninsula (supplementary Fig. 2). “

New Ms In 206 - 221.

“We searched the peer-reviewed literature for data on sediment cores dated with ²¹⁰Pb, including CaCO₃ or C_{inorg} concentration in seagrass and mangrove sediments. Search terms on Google Scholar were “seagrass” OR “mangrove” AND “²¹⁰Pb” OR “SAR” OR “sediment accretion rate”. We then searched returned articles that contained data on SAR and CaCO₃ or C_{inorg} data. We found only one study presenting CaCO₃ content in a dated sediment core. However, we found 15 and 22 studies with sediment accretion rates (SAR) for seagrass and mangrove sediments, respectively. To obtain the CaCO₃ or C_{inorg} concentrations needed to calculate C_{inorg} burial rates, we used the database of Mazarrasa et al. (2015)¹³, which was the most recent exhaustive compilation of sediment cores from blue carbon habitats, for data on CaCO₃ in seagrass sediments. We also contacted experts in blue carbon studies (published studies using cores from blue carbon habitats) for unpublished CaCO₃ sediment concentration data (see data and references in supplementary dataset). In total, we compiled 42 and 53 ²¹⁰Pb dated cores with CaCO₃ content in mangrove and seagrass ecosystems, respectively (see PRISMA checklist and flow diagram⁵⁰ in supplementary material).

The SARs (cm yr⁻¹) from the literature were re-calculated according to the constant flux - constant sedimentation model⁵¹ in order to have a coherent and comparable dating system between all cores. The

CaCO₃ concentration (% sediment dry weight) was calculated as the mean between all slices younger than 1900, for cores with the contemporary ²¹⁰Pb chronologies. The C_{inorg} concentration in sediment (gC_{inorg} m⁻³) was calculated from the dry bulk density (g m⁻³) and the percentage of CaCO₃ content (using sediment dry weight), considering a mass ratio of 12% carbon in CaCO₃. The C_{inorg} burial rate (gC_{inorg} m⁻² yr⁻¹) was then calculated as the product of the SAR and the C_{inorg} concentration for each sediment core. Cores with negligible content of CaCO₃ were also included in the calculation (see supplementary Fig. 1).

All cores from the same site or area and with similar presence or absence of allochthonous sources of CaCO₃ (see below) were treated as replicates for a global location and averaged for the analysis (geologic grouping). For seagrass, the 51 cores dated with ²¹⁰Pb were grouped into 17 locations (Fig. 2, 3). For mangroves, we compiled a total of 42 cores dated with ²¹⁰Pb in 8 locations (Fig. 2, 3). Seagrass locations ranged from tropical to sub-arctic locations, with 50% of estimates derived from tropical and subtropical locations and 50% from higher latitudes. Mangrove sediment derived mostly from subtropical locations (7 out of 8 locations), particularly in Australia and the Arabian Peninsula (supplementary Fig. 2).

Some smaller points:

Line 1: Perhaps say Calcium carbonates

We changed CaCO₃ to calcium carbonates.

Line 7: should define Allochthonous

We replaced it by “adjacent ecosystems”

Line 30: “Large” feels awkward. Perhaps “large (or high) CaCO₃ burial rates”?

We replaced “large” by “high”.

Lines 43-46: “We compared... We then address” ... need to match tenses here.

We changed the tenses to present.

Lines 71-72: Recall that to go from 90-80% CaCO₃, one needs to dissolve over half of the carbonate. Therefore, %CaCO₃ is a pretty poor measure of CaCO₃ preservation rate. This could play into the short- and long-term offsets in accumulation rate seen here.

These lines were removed as being part of the ¹⁴C, long term, results.

Lines 95-97: Recommend not using passive voice. Perhaps, “We assessed the balance between calcification, dissolution, and burial of CaCO₃ in three seagrass ecosystem locations:”

We changed the lines as proposed.

Line 102: Awkward use of “CO₂ emissions from net calcification metabolism”, especially because “net calcification metabolism” is then referred to later using the same value. Table 2 makes it clear that the actual calcification rate is different than the value given here, but this section could be made clearer.

We rewrote the section.

Old MS: 98 - 110

“The most comprehensive assessment of seagrass carbon budgets is that reported for a Mediterranean *Posidonia oceanica* meadow at Magalluf (Mallorca Island, Spain), based on flux measurements of CO₂

sequestration and net calcification rates²⁶, together with estimates of C_{org} ²⁸. In this meadow, CO_2 sequestration by net community metabolism was estimated at $8.4 \text{ gC m}^{-2} \text{ yr}^{-1}$, consistent with the C_{org} burial rate estimated independently, at $9 \pm 2 \text{ gC}_{org} \text{ m}^{-2} \text{ yr}^{-1}$ ²⁵. CO_2 emissions from net calcification metabolism were estimated at $3.6 \text{ gC m}^{-2} \text{ yr}^{-1}$ ²⁶, consistent with estimated epiphytic (gross) $CaCO_3$ production in that region²⁵ (Table 2). Calcification therefore represents an offset of 40% of the sequestration from net community production, thereby yielding a total net CO_2 sequestration of $4.8 \text{ gC m}^{-2} \text{ yr}^{-1}$. However, the C_{inorg} burial rate in this meadow, estimated here at $226 \text{ gC m}^{-2} \text{ yr}^{-1}$, is two orders of magnitude greater than the autochthonous net calcification metabolism of $3.6 \text{ gC m}^{-2} \text{ yr}^{-1}$ (Table 2). This implies that about 90% of the $CaCO_3$ burial in this seagrass meadow must be supported by allochthonous inputs. Therefore, calculation of the CO_2 sequestration by C_{org} burial offset, by comparing C_{org} and C_{inorg} burial rates or stocks, would have concluded that this meadow is a strong source of CO_2 , whereas it is a sink (as also estimated independently through air-sea flux measurements²⁹).“

New Ms ln 90 -102

“The most comprehensive assessment of seagrass carbon budgets is that reported for a Mediterranean *Posidonia oceanica* meadow at Magalluf (Mallorca Island, Spain)^{20,21,26,27}. In this meadow, Barrón et al., 2006²³, estimated a net CO_2 uptake by primary production of $8.4 \text{ gC m}^{-2} \text{ yr}^{-1}$. This estimate was corroborated by the C_{org} burial rate, estimated independently, at $9 \pm 2 \text{ gC}_{org} \text{ m}^{-2} \text{ yr}^{-1}$ ²⁸. Barrón et al., 2006²³ also estimated net calcification rates of $51 \text{ gCaCO}_3 \text{ m}^{-2} \text{ yr}^{-1}$, corresponding to $6 \text{ gC}_{inorg} \text{ m}^{-2} \text{ yr}^{-1}$. This amount of calcification would result in a CO_2 emission of $3.6 \text{ gC m}^{-2} \text{ yr}^{-1}$ (0.6 fold the net calcification¹⁴). The CO_2 emission by calcification therefore represents an offset of 40% of the CO_2 uptake from net primary production, (thereby yielding a total CO_2 sequestration of $4.8 \text{ gC m}^{-2} \text{ yr}^{-1}$ ²³). However, the C_{inorg} burial rate in this meadow is estimated here at $226 \text{ gC}_{inorg} \text{ m}^{-2} \text{ yr}^{-1}$. This is two orders of magnitude greater than the net calcification rate of $6 \text{ gC}_{inorg} \text{ m}^{-2} \text{ yr}^{-1}$ ²³(Table 2). This implies that about 90% of the $CaCO_3$ burial in this seagrass meadow must be supported by allochthonous inputs. Therefore, calculation of the CO_2 sequestration by comparing C_{org} and C_{inorg} burial rates or stocks, would have concluded that this meadow is a strong source of CO_2 , whereas estimates of calcification rates and net primary production concludes that it is a sink (as confirmed independently through air-sea flux measurements²⁷). “

Line 112: Where/how is this dissolution rate calculated? This sentence makes it sound like Table 2 shows dissolution rates when in fact this is something reported from a previous study.

Indeed, all the data comes from the article of Walker and Woelkerling, 1988, reference 22. In this article, they measured the production rate of $CaCO_3$ of the meadow by weighting the amount $CaCO_3$ on leaves, multiplied by the leaves turnover rates, and by TA anomaly technique on seagrass leaves incubations, that last allowing to assess the balance between calcification and dissolution.

New Ms line 103, We removed the dissolution rate, to avoid some confusion.

Line 127: “Fraction of $CaCO_3$ burial”

We changed sentence as proposed ln 119 new MS.

Lines 136-137: choose either “,” or “()”

We removed the “,” ln 128, new MS.

Lines 156-177: This section seems useful and is a good addition to the manuscript, but a reference should be made to the Materials and Methods section in terms of the statistical tests, and especially as to which variables are “fixed” versus “random” in the generalized linear model.

We explicated the terms “fixed” and “random” in the new MS In 158 - 164

“The presence/absence of coral reefs and lithogenic sources accounted for 36% of the variation in CaCO_3 while the random variables (study, lithology grouping, and marine province) accounted for 54% of the variation in CaCO_3 (See Materials and Methods for model description). Mangrove sediment samples showed a similar pattern to the seagrass meadows, and the presence of allochthonous sources had a marginally significant, positive effect on the amount of CaCO_3 in the sediment (t-value = 4.29, df = 1.81, $p = 0.0596$). The presence/absence of a CaCO_3 source accounted for 71% of the variation of in CaCO_3 within mangrove sediments, while the random variables accounted for 20% of the variation in CaCO_3 .”

Table 1: A reference to the Methods section would be helpful for why the “Global” and “Sum” values are slightly off from one another.

The method section explaining the table is the following:

“Calculation of global yearly burial rates of C_{inorg} ”

The global annual burial of inorganic carbon ($\text{Tg}C_{\text{inorg}} \text{ yr}^{-1}$) in seagrass meadows was calculated as the product of the global median C_{inorg} burial rates and the estimated global seagrass area, which ranges from 150,000 to 600,000 km^2 ⁹. We also calculated the global annual burial of C_{inorg} as the sum of separate calculations for tropical and arid climates and meadows at higher latitude climates. Median C_{inorg} burial rates were calculated for tropical (core locations with tropical and hot desert climates) and non-tropical areas (temperate, continental and polar climates) and multiplied by the global seagrass cover range under the assumption that 2/3 of the seagrass area is in the tropical and subtropical zone¹³. The global annual burial of inorganic carbon ($\text{Tg}C_{\text{inorg}} \text{ yr}^{-1}$) in mangroves was calculated as the product of the global median C_{inorg} burial rates and the estimated global mangrove cover of 137,760 km^2 ⁶⁷. “

We inserted a reference to that method section in the table caption as requested:

“Table 1. Median (mean) global C_{inorg} burial rates for seagrass meadows and mangrove forests considering one, and, for seagrass, two world regions (tropical and higher latitudes). See Material and Method, section “Calculation of global yearly burial rates of C_{inorg} ” for details regarding the calculations. “

Table 2: What is the difference between “Community production rate” and “Ecosystem net calcification rate”? These need to be defined in some way. For Florida Bay in particular, these two values are quite different from each other; for the others, they seem to be in better agreement.

The Community production rates are obtained by multiplying the calcifiers standing stock by the turnover rates of the species, or the leaves of the seagrass, in the case of epibionts. This estimate does not consider dissolution of carbonates, only the production. The net calcification is obtained through the measure of total alkalinity variations over time, either during incubations or in the field.

We changed the table 2 caption to clarify the difference, new MS: “Table 2. Comparison between seagrass-associated community production rate of carbonate (obtained from standing stock assessments and leaves or calcifiers turnover rates) and community net calcification rates (balance between calcification and dissolution, calculated from variations of total alkalinity) from the literature, and carbonate burial rate in three locations with carbonate-rich sediments. “

In the table, we changed the table 2 columns titles to “Community production rate of CaCO_3 ” and “Community net calcification rate”.

Supplementary Figure 2: A) needs an axis label. All plots should be better defined in the figure caption, or the appropriate Methods section should be directly referenced for how the plots were constructed.

We changed the caption to “Supplementary Figure 2. Boxplot of raw data of the paired data (vegetated habitat, yes or no) used for the GLM (A). Boxplot shows each data point (circles) with the median (line through box) and the upper and lower quartiles (box limits), while the whiskers extend to the extreme data point but no more than 1.5 times the respective quartile. The forest plot of the meta-analysis of the paired data (B) The mean effect size and associated confidence intervals are shown for each study as squares and lines and numbers on the right. The overall finding is shown at the bottom. The funnel plot of the meta-analysis (C). The funnel plot indicates potential sampling bias of meta-analysis by comparing the observed outcome (effect size) with the standard error and points outside of the white funnel indicate bias because of a high absolute value of the outcome and high standard error. See Methods for analysis details.”

We added an axis label to the panel A.